# The FAM104 proteins VCF1/2 promote the nuclear localization of p97/VCP

Maria Körner[1†], Susanne R Meyer[1†], Gabriella Marincola[1‡], Maximilian J Kern[2§], Clemens Grimm[1], Christina Schuelein-Voelk[3], Utz Fischer[1], Kay Hofmann[4], Alexander Buchberger[1*]

[1]University of Würzburg, Biocenter, Chair of Biochemistry I, Würzburg, Germany; [2]Department of Molecular Cell Biology, Max Planck Institute of Biochemistry, Martinsried, Germany; [3]Core Unit High-Content Microscopy, Biocenter, University of Würzburg, Würzburg, Germany; [4]Institute of Genetics, University of Cologne, Cologne, Germany

**Abstract** The ATPase p97 (also known as VCP, Cdc48) has crucial functions in a variety of important cellular processes such as protein quality control, organellar homeostasis, and DNA damage repair, and its de-regulation is linked to neuromuscular diseases and cancer. p97 is tightly controlled by numerous regulatory cofactors, but the full range and function of the p97–cofactor network is unknown. Here, we identify the hitherto uncharacterized FAM104 proteins as a conserved family of p97 interactors. The two human family members VCP nuclear cofactor family member 1 and 2 (VCF1/2) bind p97 directly via a novel, alpha-helical motif and associate with p97-UFD1-NPL4 and p97-UBXN2B complexes in cells. VCF1/2 localize to the nucleus and promote the nuclear import of p97. Loss of VCF1/2 results in reduced nuclear p97 levels, slow growth, and hypersensitivity to chemical inhibition of p97 in the absence and presence of DNA damage, suggesting that FAM104 proteins are critical regulators of nuclear p97 functions.

**\*For correspondence:**
alexander.buchberger@uni-wuerzburg.de

[†]These authors contributed equally to this work

**Present address:** [‡]Labor Dr. Brunner, Konstanz, Germany; [§]Roche Diagnostics GmbH, Penzberg, Germany

**Competing interest:** The authors declare that no competing interests exist.

## Editor's evaluation

This article reports on hitherto unrecognized adaptors of p97/VCP, which is a multifunctional ATPase that unwinds diverse protein substrates subserving important roles in cell physiology. The adaptors in question, members of the FAM104 family, direct p97 to the nucleus to enable unwinding events in that location. The findings, which are supported by solid experimental observations, are valuable and will inform the work of the sizable community that studies various aspects of p97/VCP.

## Introduction

The abundant, highly conserved AAA+-type ATPase p97 (also known as VCP in mammals and Cdc48 in plants and lower eukaryotes) plays a central role in the maintenance of protein and organelle homeostasis as well as genome integrity (reviewed in *Ahlstedt et al., 2022*; *Franz et al., 2016a*; *Papadopoulos and Meyer, 2017*). The best-characterized function of p97 is the segregation of ubiquitin-modified proteins in various protein quality control pathways for their subsequent degradation by the 26S proteasome (*Brandman et al., 2012*; *Dantuma and Hoppe, 2012*; *Tanaka et al., 2010*; *Ye et al., 2001*). p97 is also involved in the lysosomal degradation of endocytic cargo (*Ritz et al., 2011*) and in the selective autophagy of protein aggregates, stress granules, as well as damaged mitochondria and lysosomes (*Buchan et al., 2013*; *Ju et al., 2008*; *Kim et al., 2013*; *Papadopoulos et al., 2017*; *Tanaka et al., 2010*; *Turakhiya et al., 2018*). Moreover, p97 possesses a number of crucial nuclear functions, for instance, in DNA replication and DNA damage repair (*Acs et al., 2011*; *Davis*

*et al., 2012*; *Franz et al., 2011*; *Maric et al., 2014*; *Meerang et al., 2011*; *Moreno et al., 2014*; *Mosbech et al., 2012*; *Mouysset et al., 2008*; *Raman et al., 2011*). Importantly, mutational perturbation of p97 function causes the neuromuscular degenerative disease multisystem proteinopathy 1 (MSP1) (*Johnson et al., 2010*; *Pfeffer et al., 2022*; *Watts et al., 2004*), and several cancers as well as viral and bacterial pathogens rely on p97 activity, making p97 an attractive target for therapeutic intervention (*Das and Dudley, 2021*; *Deshaies, 2014*; *Humphreys et al., 2009*; *Huryn et al., 2020*).

The molecular basis underlying the diverse cellular functions of p97 is the partial or complete unfolding of substrate proteins by threading through the ring-shaped p97 homohexamer in an ATP hydrolysis-driven process (*Blythe et al., 2017*; *Bodnar and Rapoport, 2017*; *Buchberger, 2022*). Since p97 itself lacks appreciable specificity for its physiological substrates, a large number of cofactor proteins control substrate binding, subcellular localization, and oligomeric state of p97 (reviewed in *Buchberger et al., 2015*). Some of these cofactors bind to p97 in a mutually exclusive manner and define functionally distinct, major p97 complexes: a heterodimer of UFD1 (also known as UFD1L) and NPL4 (also known as NPLOC4) recruits ubiquitylated substrates for p97-dependent unfolding and subsequent proteasomal degradation or processing (*Bays et al., 2001*; *Jarosch et al., 2002*; *Rape et al., 2001*; *Ye et al., 2001*), whereas UBXN6 (also known as UBXD1) controls p97 functions in lysosomal and autophagic degradation pathways (*Papadopoulos et al., 2017*; *Ritz et al., 2011*) and SEP domain-containing cofactors such as p47 (also known as NSFL1C) and UBXN2B (also known as p37) mediate the maturation of protein phosphatase 1 complexes (*Weith et al., 2018*). At least in the case of p97-UFD1-NPL4, auxiliary cofactors from the UBA-UBX family can fine-tune the subcellular localization and/or substrate specificity of a major p97 complex (*Buchberger et al., 2015*; *Schuberth and Buchberger, 2008*). For example, the yeast cofactor Ubx2 recruits Cdc48-Ufd1-Npl4 to the endoplasmic reticulum (ER) and mitochondria to promote ER- and outer mitochondrial membrane-associated protein degradation, respectively (*Mårtensson et al., 2019*; *Neuber et al., 2005*; *Schuberth and Buchberger, 2005*). Similarly, the metazoan cofactors UBXN7 and FAF1 direct p97-UFD1-NPL4 to nuclear chromatin to promote the removal of ubiquitylated substrates such as CDT1 and the CMG helicase (*Franz et al., 2016b*; *Fujisawa et al., 2022*; *Kochenova et al., 2022*; *Tarcan et al., 2022*).

The majority of cofactors interact with p97 through a small number of defined domains/motifs, including the UBX(-like) domain, the PUB and PUL domains, the SHP box, the VCP interacting motif (VIM), and the VCP binding motif (VBM) (reviewed in *Buchberger et al., 2015*). However, some p97 interactors do not possess any of these canonical binding motifs, suggesting that the current inventory of p97 cofactors is incomplete and, hence, that the full scope of the p97–cofactor network is far from being understood. Examples illustrating this point include ZFAND1 (yeast Cuz1) and GIGYF1/2 (yeast Smy2), which were recently shown to regulate p97 functions in stress granule clearance (*Turakhiya et al., 2018*) and the transcription stress response (*Lehner et al., 2022*), respectively.

Here, we report the identification of the previously uncharacterized, evolutionarily conserved FAM104 protein family as p97 cofactors. We show that the human FAM104 family members VCF1 and VCF2 bind p97 directly via a non-canonical helical motif and that they associate with different p97 complexes in cells. We demonstrate that VCF1/2 promote the nuclear localization of p97 and that their loss causes impaired growth and hypersensitivity to chemical inhibition of p97.

## Results

### FAM104 proteins are a conserved family of p97 interactors

In our ongoing efforts to identify p97 binding partners, we isolated multiple clones encoding the hitherto uncharacterized proteins FAM104A (also known as FLJ14775) and FAM104B (also known as FLJ20434, CXorf44) in a yeast two-hybrid screen of a human testis cDNA library (*Supplementary file 1*). In the following, we will use the new official names VCF1 and VCF2 for human FAM104A and FAM104B, respectively, and the term FAM104 when addressing the entire protein family. VCF1 and 2 are both expressed in various isoforms originating from alternative transcript variants (*Figure 1A*). For VCF1, the two-hybrid hits matched isoforms 1, 2, and 5, whereas isoforms 3 and 4, which share just the N-terminal 73 residues with isoforms 1/2, were not isolated. For VCF2, the two-hybrid hits matched isoforms 3 and 4/5/6, respectively, which differ by the insertion of a single valine residue after residue 40. Compared to isoform 4, isoforms 5 and 6 of VCF2 possess slightly different N termini. Isoforms

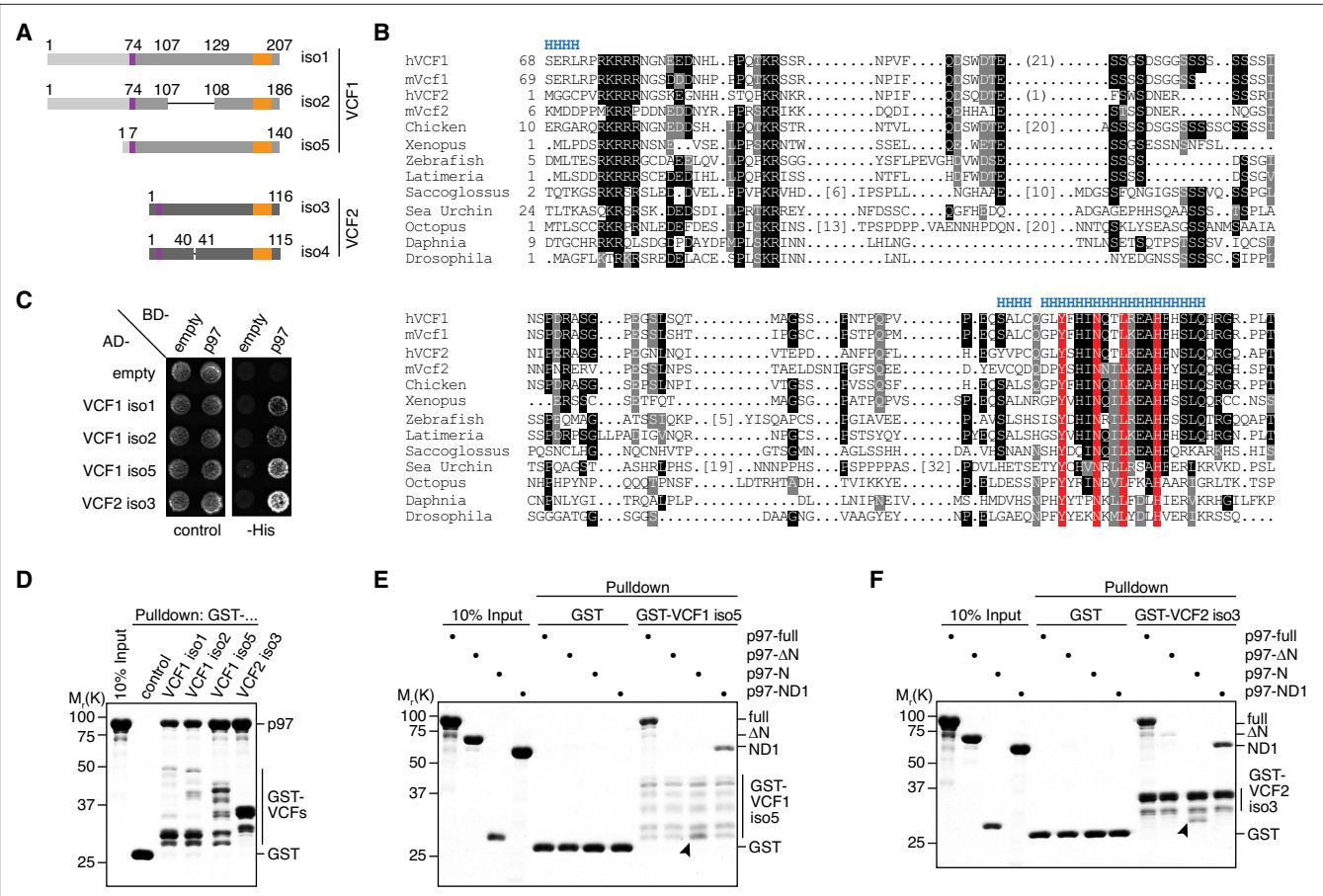

**Figure 1.** VCF1/2 bind to p97. (**A**) Schematic overview of human VCF1 and VCF2 isoforms isolated in a yeast two-hybrid screen. Relevant amino acid residue numbers are shown, and conserved N- and C-terminal sequence motifs are indicated by purple and orange boxes, respectively. Sequence identity outside these boxes is indicated by different shades of gray (light, medium, dark). Internal deletions in VCF1 isoform 2 and VCF2 isoform 4 are indicated by thin lines. (**B**) Multiple-sequence alignment showing representative members of the FAM104 family. Regions with predicted alpha-helical secondary structure are indicated at the top by 'H'. The most highly conserved residues in the C-terminal sequence motif are boxed in red. Other conserved residues are boxed in black or gray, according to the degree of conservation. Numbers in square brackets indicate the length of insertions. For human VCF1 and VCF2, the sequences of isoforms 2 and 4, respectively, are shown, and numbers in round brackets indicate a 21-residue insertion present in isoforms 1 and 5 of VCF1 and a 1-residue insertion present in isoform 3 of VCF2, respectively. (**C**) Yeast two-hybrid analysis. Yeast PJ69-4a reporter cells transformed with the indicated combinations of bait (BD-) and prey (AD-) plasmids were spotted onto agar plates containing synthetic complete medium lacking uracil and leucine (control) or uracil, leucine, and histidine (-His). Growth was monitored after 3 d. (**D**) Glutathione sepharose pulldown assay using wild-type p97 and GST fusions of the indicated VCF1/2 proteins. Binding of p97 was analyzed by SDS-PAGE followed by Coomassie brilliant blue staining. (**E, F**) Glutathione sepharose pulldown assays as in (**D**), using the indicated p97 variants and GST fusions of VCF1 isoform 5 (**E**) and VCF2 isoform 3 (**F**), respectively. Arrowheads mark the position of the p97 N domain.

The online version of this article includes the following source data and figure supplement(s) for figure 1:

**Source data 1.** Related to *Figure 1D–F*.

**Figure supplement 1.** Evolutionary conservation of FAM104 proteins.

1 and 2, which possess 30 divergent C-terminal residues, and isoform 7, which is truncated after 46 residues, were not isolated. Of note, those isoforms of VCF1 and 2 that were isolated in the two-hybrid screen possess significant sequence homology, suggesting that they are evolutionarily related. Indeed, FAM104-like proteins are present in vertebrates as well as in many invertebrates, including insects, octopuses, and echinodermata, whereas the occurrence of two distinct, VCF1- and VCF2-like homologs appears to be restricted to mammals (*Figure 1B*; *Figure 1—figure supplement 1A*). Basically, FAM104 family members possess a predicted mono- or bipartite classical nuclear localization signal (cNLS) at or close to the N terminus and a C-terminal, highly conserved helical region, which

are separated by largely unstructured stretches of variable length and lower sequence conservation (*Figure 1A and B* and *Figure 1—figure supplement 1B*).

To validate the results of the two-hybrid screen, we tested isoforms 1, 2, and 5 of VCF1 and isoform 3 of VCF2 in directed yeast two-hybrid assays and confirmed that all four full-length proteins interact with p97 (*Figure 1C*). Next, we performed glutathione sepharose pulldown experiments with purified GST fusions of these four VCF1/2 proteins and found that they were all able to efficiently bind recombinant p97 (*Figure 1D*). In order to determine the VCF1/2 binding region of p97, we performed pulldown assays with VCF1 isoform 5 and VCF2 isoform 3 using different truncated variants of p97 (*Figure 1E and F*). The ND1 variant of p97 lacking the D2 ATPase domain as well as the N domain alone bound efficiently to VCF1 and 2, whereas the ΔN variant lacking the N domain did not. Taken together, these results show that FAM104 proteins are a new family of evolutionarily conserved p97 interactors, and that the N domain is necessary and sufficient for the direct binding of p97 to VCF1/2.

## FAM104 proteins bind to p97 via their C-terminal helix

As VCF1/2 do not possess any of the canonical p97 binding motifs found in other cofactors, we next sought to map their binding site for p97. The shortest VCF1 fragment isolated in the two-hybrid screen

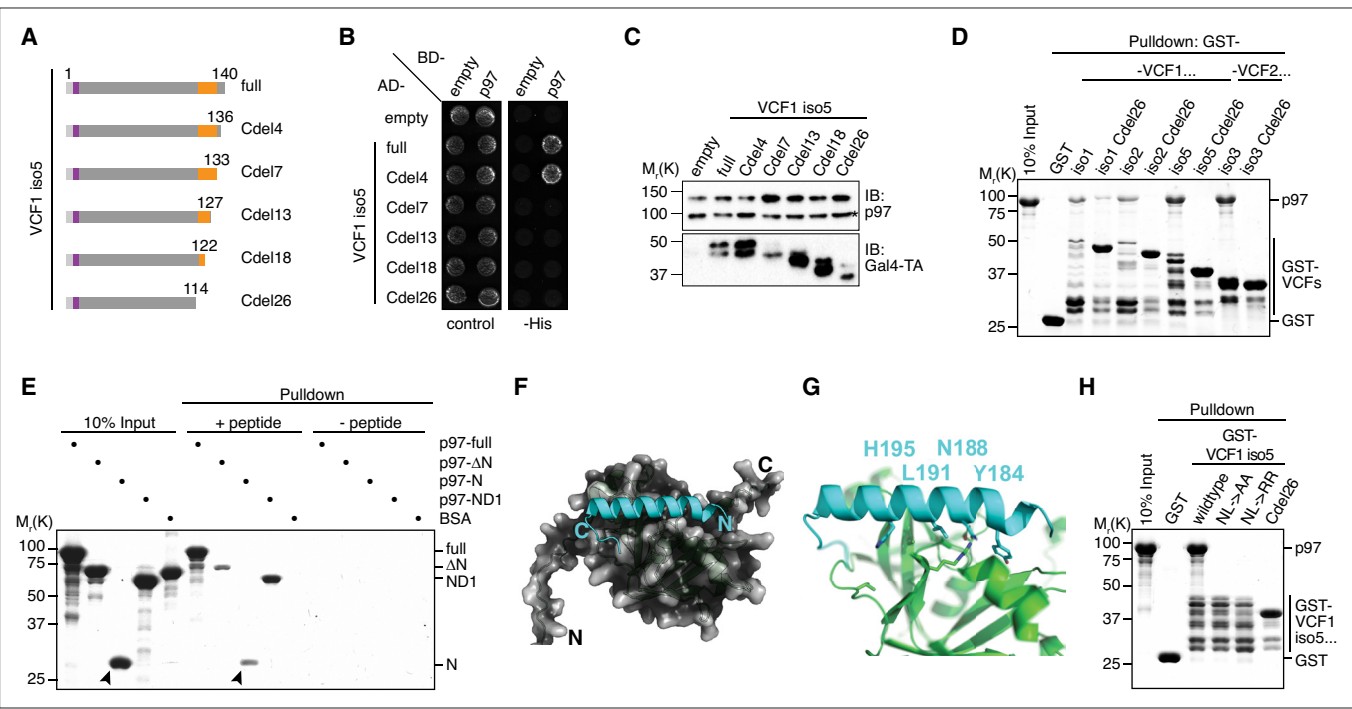

**Figure 2.** FAM104 proteins bind to p97 via their C-terminal helix. (**A**) Schematic overview of the C-terminal truncations of VCF1 isoform 5 used for yeast two-hybrid analysis. Labeling as in *Figure 1A*. (**B**) Yeast two-hybrid analysis of p97 binding to the C-terminally truncated VCF1 isoform 5 variants shown in (**A**). (**C**) Expression levels of the two-hybrid fusion proteins in (**B**) were analyzed by western blot (WB) using antibodies against p97 and the Gal4 transactivation domain (Gal4-TA). The asterisk in the p97 blot marks a cross-reactivity with endogenous Cdc48. (**D**) Glutathione sepharose pulldown assay using wild-type p97 and GST fusions of the indicated full-length or C-terminally truncated VCF1/2 proteins. (**E**) Streptavidin sepharose pulldown assay using the biotinylated peptide CQGLYFHINQTLREAHFHSLQHRG spanning the conserved C-terminal alpha-helix and flanking residues of VCF1 (residues C180–G203 in isoform 1) and the indicated p97 variants. p97 binding to the immobilized peptide was analyzed by SDS-PAGE, followed by Coomassie brilliant blue staining. Arrowheads mark the position of the p97 N domain. (**F, G**) AlphaFold Multimer model of the C-terminal alpha-helix of VCF1 (turquoise) bound to the N domain of p97. (**F**) Overview showing binding to the subdomain cleft of the N domain. (**G**) Close-up view showing the interaction of the four most highly conserved residues with the N domain (green). Residue numbers refer to isoform 1 of VCF1. (**H**) Glutathione sepharose pulldown assay using wild-type p97 and GST fusions of the indicated full-length (wildtype, NL->AA, NL->RR) or C-terminally truncated (Cdel26) variants of VCF1 isoform 5. NL->AA, N188A/L191A double mutant; NL->RR, N188R/L191R double mutant.

The online version of this article includes the following source data and figure supplement(s) for figure 2:

**Source data 1.** Related to *Figure 2C*.

**Source data 2.** Related to *Figure 2D, E, and H*.

**Figure supplement 1.** The p97 binding mode is conserved in invertebrate FAM104 proteins.

starts at glycine residue G131 of isoform 1 (equivalent to G110 and G64 of isoforms 2 and 5, respectively). Since the C-terminal alpha-helical region shows the highest sequence conservation among FAM104 family members (*Figure 1B*) and is missing in those isoforms of VCF1/2 that were not isolated in the two-hybrid screen, we truncated this region from the C-terminus to test its importance for p97 binding (*Figure 2A*). While deletion of the four C-terminal residues did not affect the two-hybrid interaction between VCF1 isoform 5 and p97, larger deletions of 7–26 residues completely abolished p97 binding (*Figure 2B*), even though the truncated proteins were expressed to similar levels in the reporter yeast strain (*Figure 2C*). Consistent with this, deletion of the C-terminal 26 residues impaired the ability of all four VCF1/2 isoforms to bind p97 in a pulldown experiment (*Figure 2D*), indicating that the C-terminal region contains the p97 binding site. We therefore tested whether a peptide spanning the conserved alpha-helical and flanking residues (residues C180 to G203 in VCF1 isoform 1) can bind to p97. To that end, we immobilized the N-terminally biotinylated peptide on streptavidin sepharose beads and performed a pulldown assay with p97 (*Figure 2E*). The peptide pulled down full-length p97 as well as the ND1 and N domain variants, demonstrating that the C-terminal conserved region of FAM104 family proteins is not only necessary, but also sufficient for p97 binding. We also noted some residual binding of the p97 ΔN variant, suggesting that the VCF1-derived peptide has some weak affinity for a p97 region(s) outside the N domain.

Using AlphaFold Multimer, we next generated a structural model of the C-terminal region of VCF1 bound to the N domain of p97 (*Figure 2FG*). The central part of the C-terminal region was predicted as an alpha-helix that binds to the subdomain cleft of the p97 N domain (*Figure 2F*), which is the major binding site for other N domain binding motifs including the UBX(-like) domain and the VIM and VBM linear motifs (*Buchberger et al., 2015*). Closer inspection of the modeled interface revealed that the four most strongly conserved residues of the helix, Y184, N188, L191, and H195, all contact the N domain (*Figure 2G*; *Figure 1B*). The predicted interactions are centered around L191, which occupies a predominantly hydrophobic pocket in the p97 subdomain cleft. Adjacent residues contribute to a hydrogen bonding network with p97, while Y184 is predicted to exhibit an orthogonal pi-stacking with Y138 of p97. To test the importance of this interface, we mutated the two central residues N188 and L191 to either alanine or arginine. We reasoned that the NL->AA double exchange should eliminate key contacts with the N domain, whereas the NL->RR double exchange should introduce steric clashes. Intriguingly, both double mutations abolished the interaction of VCF1 with p97 as efficiently as the deletion of the entire C-terminal region (*Figure 2H*). Consistent with a conserved essential role of residues N188 and L191 for p97 binding, an AlphaFold Multimer model of the *Drosophila melanogaster* homologs of VCF1/2 (CG14229) and p97 (TER94) showed a highly similar binding interface with basically identical positions of the equivalent residues N90 and L93 (*Figure 2—figure supplement 1*). Together, our data identify the C-terminal alpha-helix of FAM104 proteins as a novel p97 binding motif and show that the strictly conserved residues N188 and L191 are of central importance for the interaction.

## VCF1/2 bind to several p97 complexes in cells

To investigate whether FAM104 proteins associate with p97 in cells, we ectopically expressed N-terminally FLAG epitope-tagged VCF1/2 in HEK293T cells and performed anti-FLAG immunoprecipitations (*Figure 3A*). Endogenous p97 was efficiently co-precipitated with all four wild-type VCF1/2 proteins tested, despite the fact that VCF1 isoform 5 was less and VCF2 isoform 3 was much less well expressed than VCF1 isoforms 1 and 2, and that the levels of soluble p97 in the input were strongly reduced upon ectopic expression of VCF1 isoforms 1 and 2. Importantly, UFD1 and NPL4 were co-precipitated at levels roughly proportional to those of p97, strongly suggesting that VCP1/2 form complexes with p97-UFD1-NPL4. This conclusion is further supported by the finding that VCF1 Cdel26 variants lacking the C-terminal helix failed to co-precipitate not only p97, as expected, but also UFD1 and NPL4. (The expression level of the VCF2 isoform 3 Cdel26 variant was below the detection limit, precluding its analysis in this experiment.) We also detected a robust co-precipitation of the SEP domain cofactor UBXN2B with the wild-type, but not Cdel26 VCF1/2 proteins (*Figure 3A* and *Figure 3—figure supplement 1A and B*), indicating the existence of ternary p97-UBXN2B-VCF1/2 complexes and thereby confirming and extending recent data from high-throughput interaction studies (*Huttlin et al., 2017*; *Luck et al., 2020*). By contrast, we were unable to confirm the recently reported co-immunoprecipitation of the related SEP domain cofactor p47 with VCF1 (*Figure 3—figure supplement 1A*; *Raman*

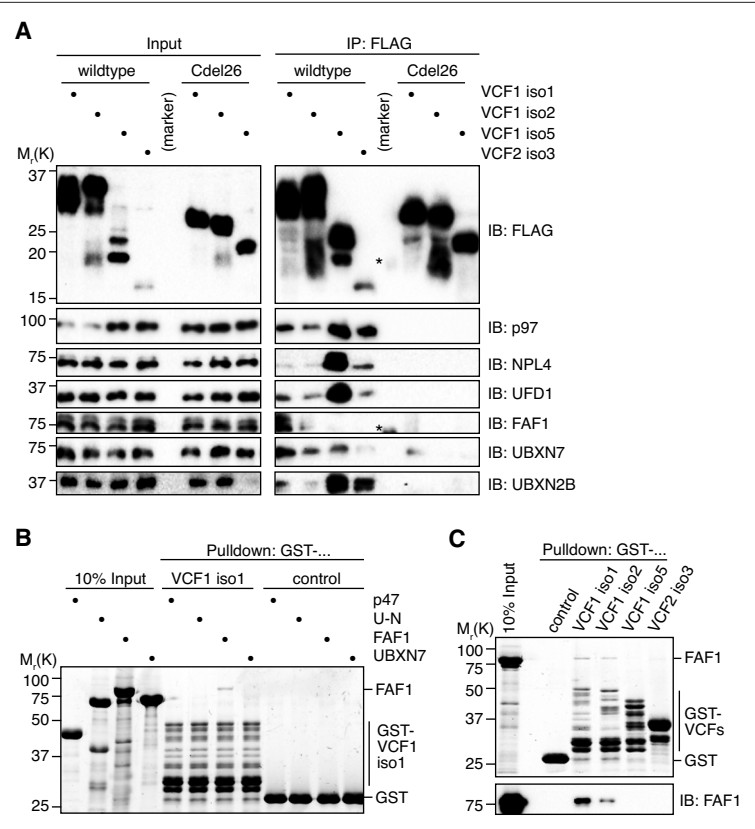

**Figure 3.** VCF1/2 form complexes with p97 and several p97 cofactors in cells. (**A**) HEK293T cells ectopically expressing the indicated N-terminally FLAG-epitope-tagged wildtype or C-terminally truncated (Cdel26) VCF1/2 proteins were subjected to anti-FLAG immunoprecipitation (IP). Input and IP samples were immunoblotted for the FLAG epitope tag, p97, and the indicated p97 cofactors. The central empty lane had been loaded with a marker. The asterisks label marker bands cross-reactive with the FLAG and FAF1 antibody, respectively. (**B**) Glutathione sepharose pulldown assay, using the indicated p97 cofactors and GST-VCF1 isoform 1; U-N, UFD1-NPL4. (**C**) Glutathione sepharose pulldown assay, using FAF1 and GST fusions of the indicated VCF1/2 proteins. Upper panel: 2,2,2-trichloroethanol-stained gel; lower panel: immunoblot with FAF1 antibody.

The online version of this article includes the following source data and figure supplement(s) for figure 3:

**Source data 1.** Related to *Figure 3A*.

**Source data 2.** Related to *Figure 3B*.

**Source data 3.** Related to *Figure 3C*.

**Figure supplement 1.** VCF1/2 associate with several p97-cofactor complexes.

**Figure supplement 1—source data 1.** Related to *Figure 3—figure supplement 1A–C*.

---

*et al., 2015*), even though our data do not exclude the possibility of a transient, conditional or weak interaction with p47. Together, our data show that FAM104 proteins can associate via their C-terminal helix with distinct cellular p97 complexes including p97-UFD1-NPL4 and p97-UBXN2B, but not with all p97 complexes.

We also determined the impact of ectopically expressing wild-type VCF1/2 on p97–cofactor complexes by immunoprecipitating endogenous p97 (*Figure 3—figure supplement 1C*). Intriguingly, the strongly expressed VCF1 isoforms 1 and 2 efficiently outcompeted the major cofactors UFD1-NPL4 and p47 as well as high molecular weight ubiquitin species presumably representing polyubiquitylated substrate proteins. By contrast, the more weakly expressed VCF1 isoform 5 and VCF2 isoform 3 did not interfere with binding of UFD1-NPL4 and p47 and even appeared to increase the association of ubiquitylated substrates with p97. These results suggest that FAM104 proteins bind very tightly to p97 and that their effect on cofactor and substrate binding depends on their expression level.

Interestingly, FAF1 and UBXN7, two auxiliary cofactors important for nuclear functions of p97-UFD1-NPL4, were also co-precipitated with VCF1/2 (*Figure 3A*). Whereas FAF1 bound to isoforms 1 and 2 of VCF1, UBXN7 bound to all four VCF1/2 isoforms. Of note, the amounts of co-precipitated FAF1 and UBXN7 did not correlate with those of p97, UFD1, and NPL4, suggesting that the interaction of FAF1 and UBXN7 with VCF1/2 may not strictly depend on the p97-UFD1-NPL4 complex. To directly address this possibility, we performed pulldown experiments with purified p97 cofactors and found that FAF1, but not p47, UFD1-NPL4, or UBXN7, bound to VCF1 isoforms 1 and 2, whereas no binding of FAF1 to VCF1 isoform 5 and VCF2 isoform 3 was detected (*Figure 3B and C*). These results suggest that the N-terminal extension shared by VCF1 isoforms 1 and 2 mediates a direct interaction with FAF1. In summary, our data indicate a complex interplay of VCF1/2 and other cofactors upon binding to p97 in living cells.

## Ectopic expression of VCF1/2 increases nuclear p97 levels

All FAM104 proteins possess a bona fide cNLS at or close to the N terminus (*Figure 1A and B* and *Figure 1—figure supplement 1B and C*). To analyze their subcellular localization, we ectopically expressed N-terminally FLAG epitope-tagged VCF1/2 in HeLa cells and performed confocal immunofluorescence microscopy using anti-FLAG antibodies (*Figure 4A*). Consistent with the very high score calculated by cNLS mapper (*Kosugi et al., 2009*), all four VCF1/2 proteins strongly accumulated in the nucleus. Intriguingly, endogenous p97 co-accumulated with VCF1/2 in the nuclei of transfected cells, as evident from the comparison with neighboring non-transfected cells or with the vector control (*Figure 4A*). We next explored the effect of deleting the C-terminal helix or the cNLS of VCF1 using C-terminally truncated isoforms 1 and 2 as well as cNLS-deleted isoforms 1, 2, and 5, respectively (*Figure 4B*). The corresponding variants of the other VCF1/2 isoforms under study were poorly expressed below the detection limit of the immunofluorescence experiments, precluding their analysis. The C-terminally truncated VCF1 Cdel26 variants localized to the nucleus but failed to effectuate a significant nuclear accumulation of p97, as expected from the loss of their p97 binding site. The cNLS-deleted VCF1 variants showed a nuclear and cytoplasmic distribution, in accordance with their small size of less than 40 kDa, and a much less pronounced nuclear accumulation of p97 (see *Figure 4—figure supplement 1A* for higher intensity images of full-length versus cNLS-deleted VCF1 isoform 1).

To quantify the microscopy results, we used CellProfiler (*McQuin et al., 2018*) for segmentation of the images into nuclei and cytoplasm (see *Figure 4—figure supplement 1B* for an example) and determined the ratio of nuclear to cytoplasmic p97 signal in control-transfected versus VCF1/2-transfected cells (*Figure 4C*). The ratio increased strongly from 2.08 ± 0.01 for control-transfected cells to up to 15.46 ± 3.21 for cells transfected with wild-type VCF1 isoform 1, reflecting the robust nuclear accumulation of p97 in the presence of VCF1/2. By contrast, no significant nuclear accumulation of p97 was observed for cells transfected with the Cdel26 or the cNLS-deleted VCF1 variants. Quantification of the FLAG signals showed that the wild-type and Cdel26 VCF1/2 proteins strongly accumulated in the nucleus, whereas the cNLS-deleted variants were detected in both cytoplasm and nucleus (*Figure 4D*). Together, these results suggest that VCF1/2 target p97 via a piggy-back mechanism to the nucleus.

## Ectopic expression of VCF1 isoforms 1 and 2 induces chromatin binding of p97

As the overexpression of VCF1 isoforms 1 and 2 had caused a noticeable reduction in the p97 input levels in the immunoprecipitation experiments described above (*Figure 3A* and *Figure 3—figure supplement 1A and C*), we hypothesized that a subpopulation of p97 may have become insoluble under these conditions. To test this possibility, we performed a biochemical fractionation of HEK293T cells ectopically expressing VCF1 isoform 1 or 2 (*Figure 5A*). Both isoforms strongly accumulated in the soluble nucleoplasmic and insoluble chromatin fractions and were only weakly detected in the cytoplasmic fraction, in full agreement with the immunofluorescence experiments. Intriguingly, the overexpression of both VCF1 isoforms did not affect the total amount of p97, as judged by the signal in SDS-denatured total cell extracts, but caused a strong increase of p97 in the chromatin fraction at the expense of the cytoplasmic pool. Because the other two VCF1/2 isoforms under study were only weakly expressed, their potential to cause similar effects could not be addressed in these

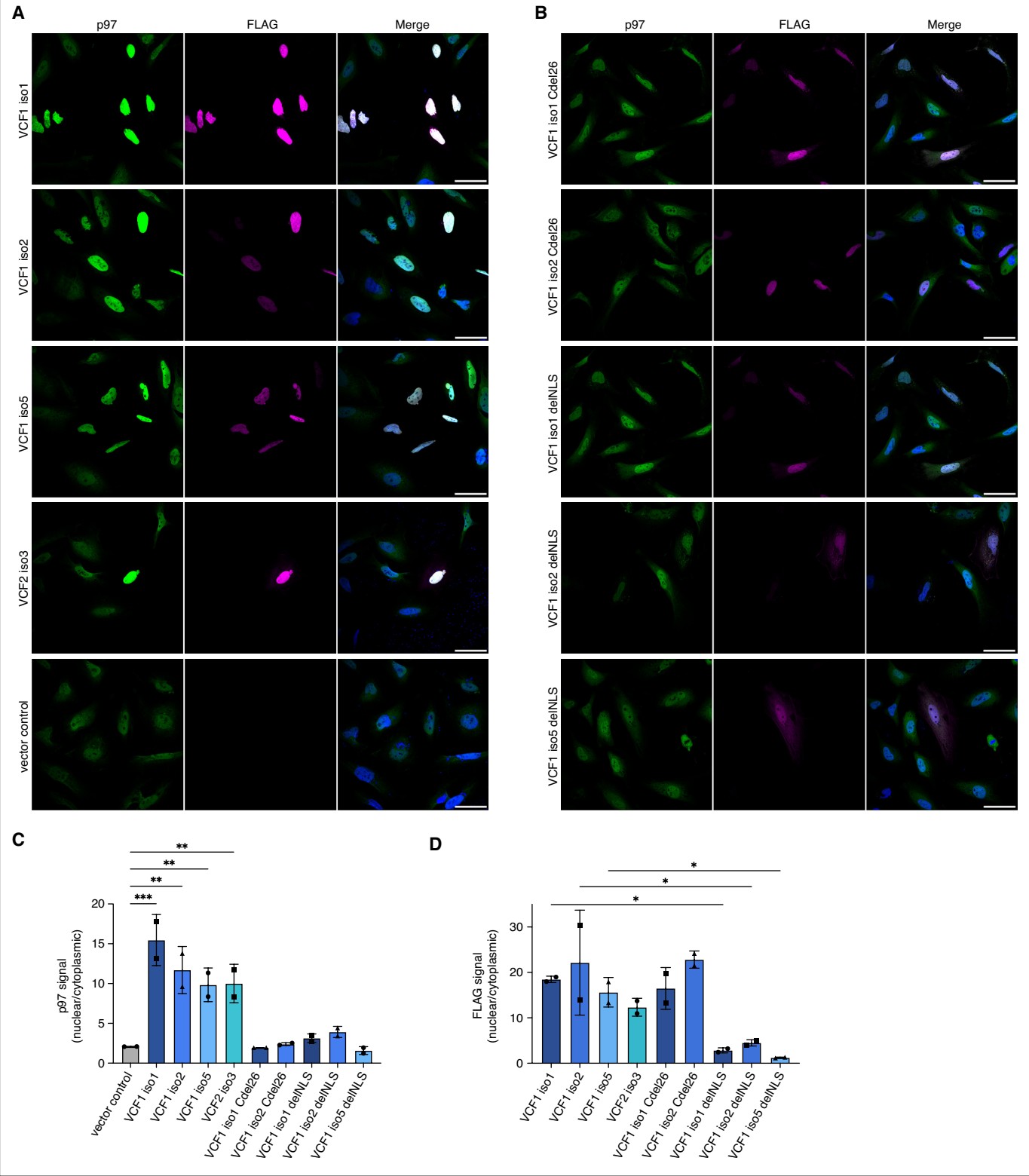

**Figure 4.** Ectopic expression of VCF1/2 increases nuclear p97 levels. (**A**) HeLa cells ectopically expressing full-length, N-terminally FLAG epitope-tagged VCF isoforms 1, 2, and 5, VCF2 isoform 3, and empty vector control, respectively, were analyzed by confocal immunofluorescence microscopy using antibodies against endogenous p97 and the FLAG epitope. Scale bars, 50 µm. All images were taken with identical acquisition settings and processed identically. (**B**) HeLa cells ectopically expressing N-terminally FLAG epitope-tagged VCF1 isoforms 1 and 2 lacking the C-terminal conserved helix (Cdel26) or the classical nuclear localization signal (cNLS) (delNLS), and VCF1 isoform 5 lacking the cNLS, were analyzed as in (**A**). Scale bars,

*Figure 4 continued on next page*

*Figure 4 continued*

50 µm. (**C**) Quantification of the ratio of nuclear to cytoplasmic p97 signals in panels (**A**) and (**B**). Except for the vector control where all imaged cells were included (>80 cells per replicate and condition), only transfected cells (as judged by the FLAG channel) were included in the quantification. Shown is the mean ± SD; n = 2 biological replicates with 15–30 transfected cells per replicate and condition; one-way ANOVA. *$p < 0.05$; **$p < 0.01$; ***$p < 0.001$; the differences between the Cdel26 and delNLS constructs and the vector control are all not significant ($p > 0.87$). (**D**) Quantification of the ratio of nuclear to cytoplasmic FLAG signals in panels (**A**) and (**B**) was performed as described in (**C**).

The online version of this article includes the following source data and figure supplement(s) for figure 4:

**Source data 1.** Related to *Figure 4C and D*.

**Figure supplement 1.** VCF1/2 promote the nuclear localization of p97.

experiments. To confirm that the reduced solubility of p97 is indeed caused by its association with chromatin, we used an alternative fractionation protocol including a benzonase treatment step to solubilize chromatin-bound proteins (*Figure 5B*). Comparing cells overexpressing VCF1 isoform 1 with control cells, we found that the majority of p97 was released into the solubilized chromatin fraction upon benzonase treatment, similar to VCF1 isoform 1 itself and the chromatin-associated proteins MYC, MCM7, and Ub-H2B.

We next analyzed the fractionation of p97 in cells expressing the Cdel26 and delNLS variants of VCF1 isoforms 1 and 2. Expression of the p97 binding-deficient Cdel26 variants did not result in any depletion of cytoplasmic p97 or accumulation of chromatin-bound p97 (*Figure 5—figure supplement 1A*), in good agreement with the microscopy data (*Figure 4BC*). Expression of the NLS-deleted

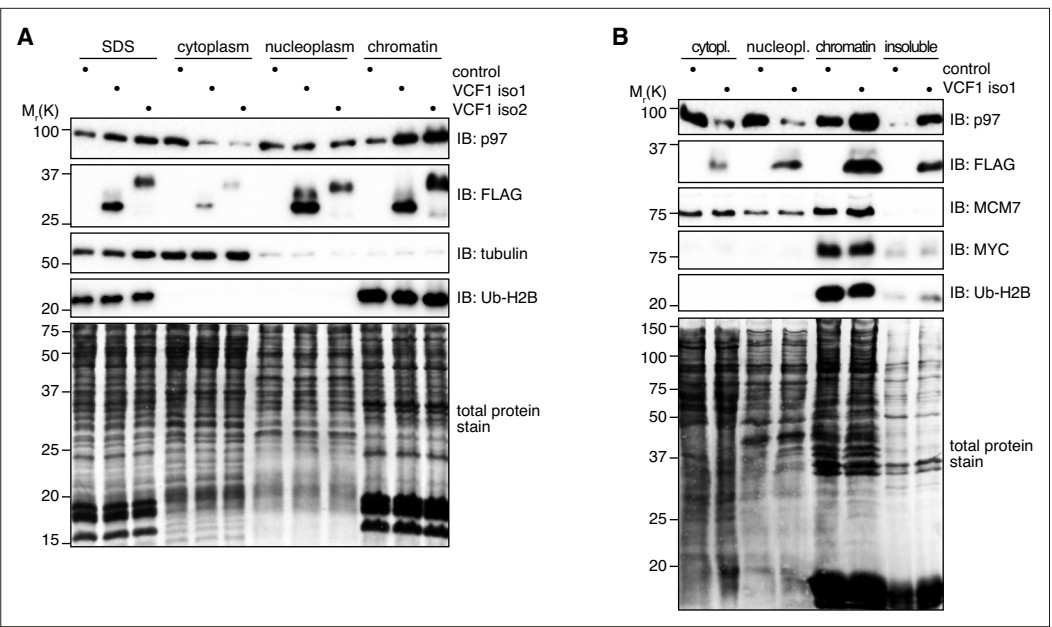

**Figure 5.** Ectopic expression of VCF1 isoforms 1 and 2 promotes the association of p97 with chromatin. (**A**) HEK293T cells were transfected with plasmids encoding the indicated N-terminally FLAG-epitope-tagged VCF1 proteins or with empty vector (control). Total protein extracts were prepared by direct boiling part of the cells in SDS-PAGE sample buffer (SDS). The remaining cells were processed to cytoplasmic, soluble nuclear (nucleoplasmic), and chromatin fractions as indicated. Tubulin and ubiquitylated histone H2B (Ub-H2B) served as markers for the cytoplasmic and chromatin fractions, respectively, whereas the Coomassie staining of the membrane (total protein stain) served as loading control. (**B**) Fractionation of lysates from HEK293T cells ectopically expressing VCF1 isoform 1 or control cells using an alternative protocol including benzonase treatment that allows to distinguish solubilized chromatin-associated proteins (chromatin) from insoluble proteins (insoluble).

The online version of this article includes the following source data and figure supplement(s) for figure 5:

**Source data 1.** Related to *Figure 5A and B*.

**Figure supplement 1.** Control fractionations of cells ectopically expressing Cdel26 and delNLS variants of VCF1 isoforms 1 and 2.

**Figure supplement 1—source data 1.** Related to *Figure 5—figure supplement 1A and B*.

variants resulted in some accumulation of p97 in the nuclear fractions, consistent with the nuclear/chromatin localization of these VCF1 isoforms (*Figure 5—figure supplement 1B*).

Taken together, our data indicate that VCF1 isoforms 1 and 2 strongly associate with chromatin, at least upon overexpression, thereby inducing chromatin binding of p97, and they provide a rationale for the reduction of soluble p97 observed in the immunoprecipitation experiments.

## Loss of VCF1/2 reduces nuclear p97 levels

Since the previous experiments relied on the ectopic expression of VCF1/2, we next sought to determine whether endogenous VCF1/2 also has an impact on the nuclear localization of p97. To this end, we generated VCF1 knockout as well as VCF1/2 double-knockout HeLa cell pools by CRISPR/Cas9-mediated genome editing. Compared to control cells, both knockout cell pools showed a significantly reduced nuclear p97 signal, with the ratio of nuclear to cytoplasmic p97 decreased by about 22% (*Figure 6A and C*). The siRNA-mediated depletion of VCF1 had a similar effect (*Figure 6B and D*), which was more obvious to the naked eye in maximum intensity projections of z stacks (*Figure 6—figure supplement 1A*). Moreover, a comparable reduction in nuclear p97 was observed in VCF1 single and VCF1/2 double-knockout HEK293T cell pools (*Figure 6—figure supplement 1B and C*). To further confirm these results, we used high-content microscopy of control and VCF1/2 double-knockout HeLa cells and found a statistically highly significant reduction of the ratio of nuclear to cytoplasmic p97 by 25% (*Figure 6—figure supplement 1D*), in excellent agreement with the confocal microscopy analyses.

Because the nuclear recruitment of p97 depended on both, the cNLS and the p97 binding helix of VCF1 (*Figure 4*), we next sought to test whether the reduction in nuclear p97 signal observed upon deletion of VCF1/2 can be compensated by providing p97 with an efficient cNLS in *cis*. To that end, we generated stable cell pools expressing FLAG epitope-tagged wild-type p97 or p97 carrying an N-terminal fusion of the prototypical SV40 cNLS under the control of a doxycycline-inducible promoter. Next, we transfected these cells with siRNAs targeting VCF1 or control siRNAs and performed high-content microscopy to determine the effect of ectopic p97 expression on nuclear p97 levels (*Figure 6E and F*). In the absence of doxycycline and in the vector control, VCF1 depletion resulted in a significant reduction of the nuclear to cytoplasmic p97 ratio by about 25%, as expected from the de(p)letion experiments described above. Intriguingly, the doxycycline-induced expression of NLS-FLAG-p97 increased the nuclear p97 signal to the same levels in the VCF1-depleted and control-depleted cells, thereby fully compensating for the lack of VCF1.

In summary, our data show that endogenous VCF1/2 modulate the nucleo-cytoplasmic distribution of p97, and that the presence of a strong cNLS at p97 either in *trans* (via binding of VCF1/2) or in *cis* (via an N-terminal fusion) is necessary for the efficient nuclear localization of p97.

## VCF1/2 are required for normal cell growth

p97 functions in a number of important nuclear processes. To analyze whether the reduction of nuclear p97 in the absence of VCF1/2 has consequences for the cell fitness, we performed growth assays with HeLa VCF1/2 double-knockout cells (*Figure 7*). Eight days after seeding equal numbers of cells, the number of knockout cells was reduced by more than 40% compared to control cells, indicating that VCF1/2 are required for normal growth under otherwise unperturbed conditions (*Figure 7A*). Next, we determined the growth of VCF1/2 double-knockout cells under conditions of limited p97 activity, that is, in the permanent presence of low, sublethal concentrations of the ATP-competitive p97 inhibitor CB-5083 (*Anderson et al., 2015*), for 7 d. While no statistically significant growth difference between knockout and control cells was observed at lower CB-5083 concentrations, growth of the knockout cells was significantly impaired in the presence of 100 and 200 nM CB-5083, indicating that loss of VCF1/2 made the cells hypersensitive to p97 inhibition (*Figure 7B*). Note that growth of both cell lines was normalized to their respective growth in the absence of CB-5083, thus taking account of the slower growth of the knockout cells under unperturbed conditions. We also tested the effect of VCF1 overexpression by transiently transfecting HEK293T cells with the strongly expressed isoform 1 and observed slightly reduced growth and enhanced CB-5083 sensitivity (*Figure 7—figure supplement 1*). This negative effect on cellular fitness is consistent with the strong impact of VCF1 isoform1 overexpression on cofactor binding to p97 (*Figure 3—figure supplement 1C*), suggesting that a balanced expression of VCF1/2 is crucial for optimal p97 function.

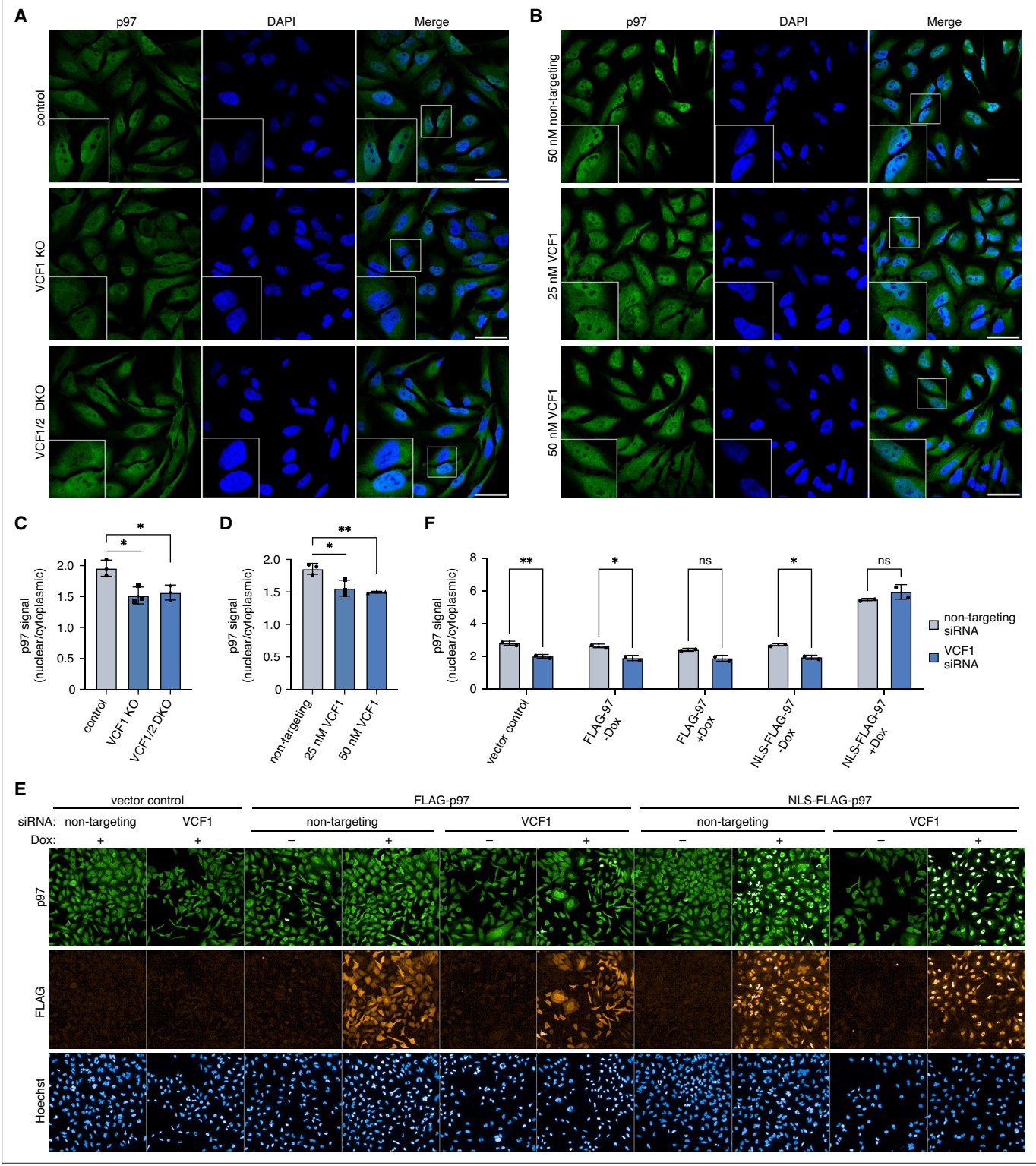

**Figure 6.** Deletion of VCF1/2 reduces nuclear p97 levels. (**A**) Control, VCF1 single-knockout (KO) and VCF1/2 double-knockout (DKO) HeLa cell pools were analyzed by confocal immunofluorescence microscopy using an antibody against endogenous p97. Scale bars, 50 μm. (**B**) HeLa cells were transfected with non-targeting or VCF1-specific siRNAs at the indicated final concentrations for 72 hr and analyzed as in (**A**). Scale bars, 50 μm. (**C, D**) Quantification of the ratios of nuclear to cytoplasmic p97 signals in (**A**) and (**B**), respectively. Shown is the mean ± SD; n = 3 biological with ≥80 cells per replicate and condition; *p<0.05; **p<0.01; unpaired, two-tailed Student's *t*-test. (**E**) HeLa cell pools expressing wild-type p97 (FLAG-p97) or p97 carrying an N-terminal fusion of the SV40 cNLS (NLS-FLAG-p97) under the control of a doxycycline-inducible promoter, or empty vector control cells,

*Figure 6 continued on next page*

*Figure 6 continued*

were transfected with VCF1-targeting or non-targeting siRNAs, induced with doxycycline (Dox) for 40 hr where indicated, stained with antibodies detecting endogenous p97 or FLAG, and subjected to high-content microscopy, followed by automated image analysis of nuclear to cytoplasmic p97 signals. (**F**) Quantification of the ratios of nuclear to cytoplasmic p97 signals in (**E**); n = 2 biological replicates (performed in three technical replicates each) with ≥3000 cells per biological replicate and cell pool, shown are mean ± SD; two-way ANOVA; *p<0.05; **p<0.01; ns, not significant.

The online version of this article includes the following source data and figure supplement(s) for figure 6:

**Source data 1.** Related to *Figure 6C, D, and F*.

**Figure supplement 1.** De(p)letion of VCF1/2 reduces nuclear p97 levels.

**Figure supplement 1—source data 1.** Related to *Figure 6—figure supplement 1C and D*.

Considering the involvement of p97 in various DNA damage repair pathways, we next analyzed the growth of VCF1/2 double-knockout cells in the presence and absence of CB-5083 after pretreatment with the DNA damage-inducing topoisomerase I inhibitor, camptothecin (*Figure 7C*). In the absence of CB-5083, the relative growth of camptothecin-treated control and knockout cells was comparable, indicating that loss of VCF1/2 alone does not cause a hypersensitivity to camptothecin. By contrast, a clear trend towards reduced growth was observed for the knockout cells in the presence of CB-5083. While this effect did not reach statistical significance at all CB-5083 concentrations, it did so at 25 nM, that is, at a concentration that had no effect in the absence of camptothecin pretreatment (see *Figure 7B*). Together, these results show that VCF1/2 are required for optimal cell fitness, particularly under conditions of limited p97 activity.

To obtain further insights into the observed hypersensitivity of VCF1/2 knockout cells to CB-5083, we determined the *VCF1* mRNA levels of wild-type HeLa cells treated with CB-5083 for 24 hr by qPCR using primer pairs specifically amplifying isoforms 1 and 2 (*Figure 7D*). The expression of *VCF1* was induced by 1.75- to 2-fold upon CB-5083 treatment, suggesting that cells try to compensate impaired p97 function by increased VCF1 levels. Interestingly, our qPCR analysis of untreated VCF1/2 single- and double-knockout cell pools revealed a maximal reduction of *VCF1* isoform 1 and 2 mRNA of 50% (*Figure 7—figure supplement 2*), potentially indicating that the homozygous deletion of VCF1 precludes cell survival and suggesting that the observed defects of VCF1/2 knockout cells may underestimate the true physiological relevance of VCF1/2.

Finally, we analyzed the impact of inhibiting p97 on its subcellular localization in control and VCF1/2 double-knockout cell pools (*Figure 7EF*). Intriguingly, treatment of control cells with 1 μM CB-5083 for 6 hr or with 0.5 μM CB-5083 for 24 hr led to an accumulation of small cytoplasmic p97-positive puncta and to a significant decrease in the ratio of nuclear to cytoplasmic p97 signal to 79 and 72%, respectively. Importantly, CB-5083 treatment decreased the already reduced nuclear p97 signal of the VCF1/2 knockout cells even further to 67 and 65%, demonstrating that the effects of p97 inhibition and lack of VCF1/2 on the nuclear localization of p97 are additive. Together with the synthetic growth defects described above, these results strongly suggest that the combined VCF1/2 deletion and inhibitor treatment reduces nuclear p97 levels and functions to an extent that is no longer compatible with viability.

## Discussion

In this study, we characterized FAM104 proteins as a family of evolutionarily conserved p97 interactors and identified their C-terminal helix as a novel p97 binding motif. We showed that the human FAM104 proteins VCF1 and VCF2 associate with p97 complexes containing the cofactors UFD-NPL4, FAF1, UBXN7, and UBXN2B in living cells, and that they efficiently recruit p97 to the nucleus. Loss of VCF1/2 reduced nuclear p97 levels, impaired cell growth, and caused hypersensitivity to p97 inhibition in the absence and presence of DNA damage, indicating that VCF1/2 support optimal p97 activity in the nuclear compartment.

Whereas VCF2 had not been reported to interact with p97 before, VCF1 (FAM104A) was identified as a p97 interactor in several high-throughput proteomic and yeast two-hybrid screening projects (*Fogeron et al., 2013*; *Haenig et al., 2020*; *Hein et al., 2015*; *Huttlin et al., 2017*; *Luck et al., 2020*; *Rolland et al., 2014*). However, the interaction between p97 and VCF1 was not further validated in any of these previous studies. In fact, the validation of p97 binding is complicated by the existence

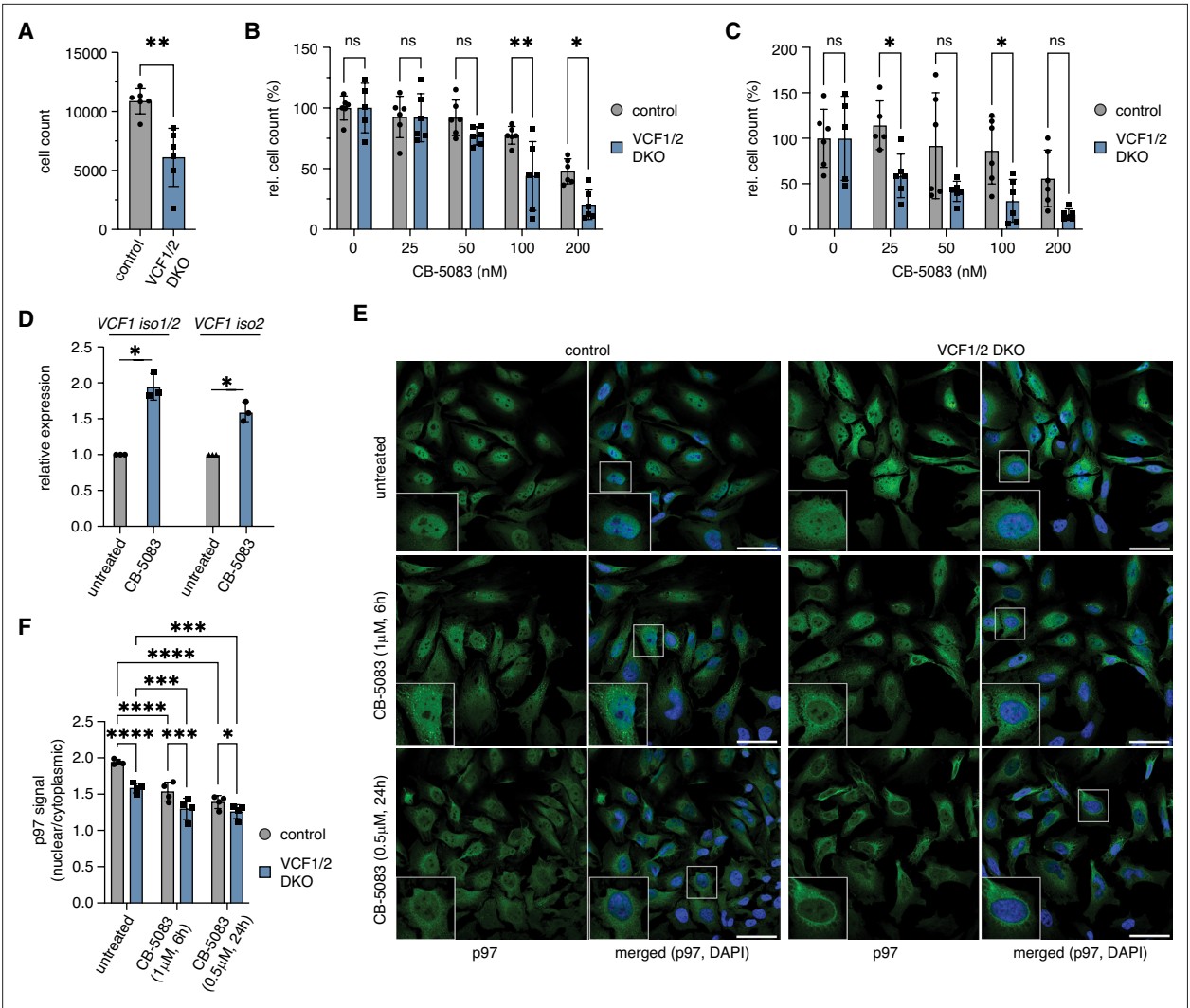

**Figure 7.** Deletion of VCF1/2 causes reduced growth and hypersensitivity to p97 inhibition. (**A**) HeLa VCF1/2 double-knockout (DKO) and control cells were seeded in 96-well plates, cultivated for 8 d, stained with Hoechst dye and automatically counted using a high-content microscopy platform. n = 6 (two biological replicates in three technical replicates each), shown are mean ± SD; unpaired two-tailed Student´s *t*-test, \*\*p<0.01. (**B**) Relative growth after 8 d was determined at the indicated concentrations of the p97 inhibitor CB-5083. Values were normalized separately for control and knockout cells to the respective mean value at 0 nM CB-5083. A two-way ANOVA was performed using the normalized data. n = 6 (two biological replicates in three technical replicates each); \*p<0.05; \*\*p<0.01; ns, not significant. (**C**) Same as in (**B**), but cells were pretreated with 1 uM camptothecin for 70 min before starting the CB-5083 treatment. (**D**) *VCF1* mRNA levels of HeLa cells grown in the presence or absence of 0.8 μM CB-5083 for 24 hr were determined by qRT-PCR using primer pairs specifically amplifying isoforms 1 and 2 or isoform 2 only, respectively. n = 3 biological replicates, shown are mean ± SD; two-tailed, one-sample *t*-test; \*p<0.05. (**E**) Control and VCF1/2 DKO HeLa cell pools were treated with CB-5083 as indicated or left untreated, followed by confocal immunofluorescence microscopy, using an antibody against endogenous p97. Scale bars, 50 μm. (**F**) Quantification of the ratios of nuclear to cytoplasmic p97 signals in (**E**); n = 4, shown are mean ± SD; two-way repeated-measures ANOVA; \*p<0.05; \*\*p<0.01; \*\*\*p<0.001; \*\*\*\*p<0.0001.

The online version of this article includes the following source data and figure supplement(s) for figure 7:

**Source data 1.** Related to *Figure 7A–D and F*.

**Figure supplement 1.** Overexpression of VCF1 isoform 1 causes moderate growth defects.

**Figure supplement 1—source data 1.** Related to *Figure 7—figure supplement 1*.

**Figure supplement 2.** VCF1 expression levels in VCF1 and VCF1/2 knockout cell pools.

**Figure supplement 2—source data 1.** Related to *Figure 7—figure supplement 2*.

of five different isoforms, of which only three (isoforms 1, 2, and 5) contain the p97-interacting C-terminal helix. The existence of p97 binding and non-binding isoforms of VCF1 and 2 in humans also initially obscured the prevalence of FAM104 proteins in the animal kingdom. Nevertheless, bioinformatic searches using the C-terminal p97 binding helix as query readily identified homologs in all five vertebrate classes as well as in *Saccoglossus* and sea urchin, leading to the subsequent identification of numerous invertebrate homologs (*Figure 1—figure supplement 1A*). Outside of the chordates, FAM104 proteins are found in many arthropods (insects, spiders, crustaceans) and selected other invertebrate lineages, including mollusks (octopi, snails), tardigrades, and rotifers, but are absent in nematodes, protists, plants, and fungi. The three human p97-interacting VCF1 isoforms possess an N-terminal 74-residue extension exclusively found in eutheria, a 21-residue insertion after residue 107 (numbering referring to isoform 1/2) that is found in mammals, birds, and reptiles, or both (*Figure 1A and B*). Amphibian, fish, and invertebrate homologs contain neither insertion and are therefore more similar to human VCF2 with respect to their overall organization. The widespread occurrence of FAM104 proteins strongly suggests a conserved function, even though very little information is available for non-human homologs. The fly homolog CG14229 lacks any functional annotations, whereas mouse Vcf1 (Fam104a) was reported to interact with ubiquitin (*Zhang et al., 2017*). The binding preference for K6-, K48-, and K29-linked di-ubiquitin observed in that study closely resembled that of p97, suggesting that Vcf1 did not bind ubiquitin directly, but in the context of a p97–cofactor complex such as p97-Ufd1-Npl4.

Here, we showed for the first time that FAM104 proteins and p97 interact directly via their highly conserved C-terminal alpha-helix and N domain, respectively (*Figures 1 and 2*). Structural modeling revealed that the C-terminal helix binds to the subdomain cleft of the p97 N domain. Interestingly, the position of the VCF1 helix in the model is similar to those of the helical VIM and VBM p97 binding motifs in the absence of any significant sequence similarity (*Buchberger et al., 2015*; *Hänzelmann and Schindelin, 2011*; *Lim et al., 2016*). Even the respective key p97-contacting residues are distinct, with N188 and L191 in the C-terminal helix of VCF1 versus predominantly arginine residues in VIM and VBM. The finding that alpha-helices can efficiently bind to the N subdomain cleft via just one or two key residues underlines the plasticity of this binding interface and raises the intriguing possibility that additional interactors bind p97 through yet other helical motifs.

Our cell-based experiments revealed a significant effect of VCF1/2 on the subcellular localization of p97 (*Figures 4–6*). Their overexpression resulted in a strong increase in p97 nuclear localization and, for VCF1 isoforms 1 and 2, chromatin association, whereas both CRISPR/Cas9-mediated deletion and siRNA-mediated depletion caused a decrease in nuclear p97 levels. While we were unable to detect endogenous VCF1/2 due to the lack of suitably sensitive antibodies, the highly consistent effects of the deletion and depletion experiments on nuclear p97 levels provide strong indirect evidence that endogenous VCF1/2 indeed localize to the nucleus, in agreement with their potent cNLS and the robust nuclear accumulation of ectopically expressed VCF1/2. Because the de(p)letion of VCF1 alone had similar effects as the VCF1/2 double knockout in these experiments, VCF1 apparently plays a more important role for the nuclear localization of p97 than VCF2, at least in HeLa and HEK293T cells.

p97 homologs from yeast to mammals possess a relatively weak potential cNLS (residues K60-R65 in human p97, score 5 according to cNLS mapper). Available three-dimensional structures of p97 suggest that this potential cNLS is part of a beta-hairpin in the N domain that is unlikely to be constitutively exposed due to nucleotide-dependent conformational changes of and cofactor binding to the N domain. Interestingly, an additional, functional bi-partite cNLS found closer to the N terminus in yeast Cdc48 (*Madeo et al., 1998*) is conserved in several p97 homologs from organisms lacking FAM104 proteins, but lost or poorly conserved in most p97 homologs from organisms possessing FAM104 proteins (*Figure 1—figure supplement 1C*). This raises the intriguing possibility that FAM104 proteins compensate the loss of the functional bipartite cNLS in the latter p97 homologs by exposing a very strong cNLS that is connected via a long, flexible region with the C-terminal helix tightly interacting with the p97 N domain. This speculation is further supported by our finding that the fusion of the strong SV40 cNLS to the N terminus of p97 can fully compensate for the loss of VCF1/2 with respect to p97 nuclear localization (*Figure 6E and F*).

On a more general note, our study underscores the importance of a sufficiently high number of NLSs that has to be present on large cargoes for their efficient nuclear import. In a recent quantitative study, a model cargo of roughly the same size as the p97 hexamer, MS2$^{S37P}$, had to

be decorated with at least 10 NLSs for detectable nuclear import (*Paci et al., 2020*), suggesting that the six moderately strong and potentially masked cNLSs in the p97 hexamer are insufficient and have to be assisted by the strong cNLSs of VCF1/2. Since de(p)letion of VCF1/2 caused only a partial reduction in nuclear p97 levels in our experiments, it is likely that additional nuclear cofactors such as UBXN7 with its strong cNLS contribute to the efficient nuclear targeting of p97. Intriguingly, two other small proteins possessing strong NLSs were recently found to possess functions analogous to VCF1/2 in the nuclear import of essential protein complexes of the ubiquitin proteasome system. AKIRIN2 and HAPSTR1 promote the nuclear import of the assembled 20S proteasome (*de Almeida et al., 2021*) and of the E3 ligase HUWE1 (*Monda et al., 2023*), respectively, indicating that the decoration of protein complexes with small, NLS-containing interaction partners is a general strategy of cells.

We were able to show that VCF1 and 2 associate with the major p97-UFD1-NPL4 complex (*Figure 3*), potentially implicating FAM104 proteins in nuclear p97 functions such as DNA damage repair, regulation of DNA replication, and control of transcription. Consistent with this possibility, isoforms 1 and 2 of VCF1 bound the auxiliary cofactors UBXN7 and FAF1 (*Figure 3*) and promoted the chromatin association of p97 upon overexpression (*Figure 5*). The finding that VCF1/2 can also associate with the p97-UBXN2B complex involved in protein phosphatase 1 regulation (*Figure 3A* and *Figure 3—figure supplement 1A*) is intriguing as it indicates that FAM104 family proteins are likely to control p97 functions beyond proteasomal degradation pathways. While it is possible that the strong chromatin association of p97 is a consequence of overexpressing the chromatin-binding VCF1 isoforms 1 and 2 and/or of the massive loss of cofactor interactions observed under this condition (*Figure 3*), the impact of VCF1/2 on p97 nuclear levels was also observed in immunofluorescence microscopy upon the much weaker ectopic expression of VCF1 isoform 5 and VCF2 isoform 3 preserving p97-cofactor complexes, as well as upon de(p)letion of VCF1/2. Thus, the accumulated evidence strongly supports a direct regulatory function of VCF1/2 on the nuclear localization of p97. It is hardly compatible with the scenario that the observed relocalizations of p97 are merely secondary, compensatory effects of increased or reduced VCF1/2 levels, respectively.

While the finding that VCF1/2 double-knockout cells are hypersensitive to p97 inhibition in the absence and presence of the DNA damaging agent camptothecin (*Figure 7B and C*) is in line with an involvement of VCF1/2 in p97-dependent DNA damage repair processes, we were unable to detect a camptothecin sensitivity in the absence of p97 inhibitor. The partial reduction of VCF1 expression by not more than 50% in the knockout cell pools (*Figure 7—figure supplement 2*) is likely to prevent the detection of more severe phenotypes. It is conceivable that the decrease in nuclear p97 levels by 25% under these conditions is insufficient to severely compromise p97-dependent nuclear processes and/or that these essential processes are safeguarded by redundant p97 cofactors. Importantly, we found that chemical inhibition reduces nuclear p97 levels and further aggravates the effect of VCF1/2 deletion on the nuclear localization of p97 (*Figure 7E and F*), strongly suggesting that the cellular basis for the synthetic growth defect observed upon combined VCF1/2 deletion and inhibitor treatment are limiting nuclear levels and activities of p97.

The nuclear recruitment function of FAM104 proteins is reminiscent of two other small p97 interactors, SVIP and VIMP (also known as SELENOS, selenoprotein S). SVIP is myristoyl-anchored to the lysosomal membrane, whereas VIMP is a single-spanning ER membrane protein. It therefore appears that FAM104 proteins, SVIP, and VIMP recruit p97 to distinct subcellular localizations with high demand for p97 activity.

According to the Human Protein Atlas and Proteomics DB databases, VCF1 and 2 are broadly expressed in many human tissues, including strong expression in the brain. In this context, it is interesting to note that both proteins have recently been implicated in neurodegenerative diseases (VCF1: amyotrophic lateral sclerosis; VCF2: Alzheimer's disease), based on systematic yeast two-hybrid screening and computationally predicted neurodegenerative disease-associated protein clusters (*Haenig et al., 2020*). Moreover, the potential link between VCF1/2 and p97-mediated DNA damage repair suggested by our data could hint to a role of VCF1/2 in p97-dependent cancer cell survival, an area of intense biomedical research efforts. While the exact role of VCF1/2 in the control of nuclear p97 functions remains to be determined, this work provides a starting point for the future exploration of the FAM104 protein family in basic and translational research.

## Materials and methods

Plasmids, antibodies, and other key resources used in this study are listed in Appendix 1—key resources table.

### Plasmids

Full-length and C-terminally truncated coding sequences of human VCF1 isoforms 1 (NCBI reference sequence NM_001098832.2), 2 (NM_032837.3), and 5 (NM_001289412.2) and of human VCF2 isoform 3 (NM_001166700.2) were PCR-amplified from a yeast two-hybrid human testis cDNA library (Clontech, Cat# 638810) and cloned via appropriate restriction sites into pGAD-C1 (*James et al., 1996*), pGEX-4T1 (Cytiva), and pCMV-Tag2B (Agilent) using standard procedures. The coding sequence of VCF1 isoform 5 was mutated using the QuikChange XL II kit (Agilent) with primers FAM104A_NL-RR_fwd (5'-CCT CTA CTT CCA CAT CCG CCA GAC CCG CAG GGA GGC CCA CTT CC), FAM104A_NL-RR_rev (5'- GGA AGT GGG CCT CCC TGC GGG TCT GGC GGA TGT GGA AGT AGA GG), FAM104A_NL-AA_fwd (5'-CCT CTA CTT CCA CAT CGC CCA GAC CGC GAG GGA GGC CCA CTT CC), and FAM104A_NL-AA_rev (5'-GGA AGT GGG CCT CCC TCG CGG TCT GGG CGA TGT GGA AGT AGA GG) according to the manufacturer's instructions. The cNLS-deleted variants of VCF1 isoforms 1 and 2 (deletion of codons 74–82) and VCF1 isoform 5 (deletion of codons 2–15) were generated by two- and one-step PCR reactions, respectively. The coding sequence of human p97 was cloned into pGBDU-C1 (*James et al., 1996*), and the coding sequence of human p97 including an N-terminal FLAG epitope tag was cloned into the lentiviral vector pINDUCER20 (Addgene plasmid #44012; gift from Stephen Elledge) (*Meerbrey et al., 2011*). To fuse the sequence encoding the SV40-derived cNLS (MPKKKRKVGGG) to the N-terminus of FLAG-p97, the QuikChange XL II kit was used according to the manufacturer's instructions with primers pIND_Nterm_SV40-NLS_fwd (5'-GAA TTC GCG GCC GCC ACC ATG CCC AAG AAA AAG CGG AAG GTG GGA GGC GGA GAT TAC AAG GAT GAC GAC G) and pIND_Nterm_SV40-NLS_rev (5'-CGT CGT CAT CCT TGT AAT CTC CGC CTC CCA CCT TCC GCT TTT CT TGG GCA TGG TGG CGG CCG CGA ATT C). The coding sequence of human UBXN7 was cloned into mini-pRSETA (*Perrett et al., 1999*). The identities of all inserts were confirmed by Sanger sequencing. Plasmids for the bacterial expression of FAF1 (*Jensen et al., 2001*), p47 (*Allen et al., 2006*), UFD1-NPL4, and full-length and truncated p97 (*Fernández-Sáiz and Buchberger, 2010*; *Rothballer et al., 2007*) have been described.

### Proteins and peptides

Expression of GST-tagged VCF1/2 proteins or of hexahistidine-tagged UBXN7 in *Escherichia coli* BL21(DE3)pRIL (Agilent) was induced with 1 mM IPTG during overnight growth at 18°C. GST fusion proteins were affinity-purified on a glutathione sepharose matrix (Cytiva) according to the manufacturer's instructions. Purified GST fusion proteins were dialyzed against TBS (50 mM Tris-HCl, 150 mM NaCl, pH 7.5) containing 2 mM DTT, aliquoted, flash frozen in liquid nitrogen and stored at –80°C. His$_6$-UBXN7 was purified by Ni$^{2+}$-NTA affinity chromatography (QIAGEN) according to the manufacturer's instructions, followed by anion exchange chromatography on a ResourceQ column (Cytiva). Purified His$_6$-UBXN7 was dialyzed against 50 mM Tris-HCl, 50 mM NaCl, 2 mM DTT, pH 8.0 and flash frozen. His$_6$-FAF1, His$_6$-p47 (*Allen et al., 2006*) and UFD1-NPL4, full-length and truncated p97 (*Fernández-Sáiz and Buchberger, 2010*) were purified as previously described.

### Yeast two-hybrid assays

The reporter yeast strain PJ69-4a (*James et al., 1996*) was transformed with combinations of bait (pGBDU) and prey (pGAD) plasmids and plated onto medium lacking uracil and leucine. Several transformant colonies were picked and resuspended in sterile water. Cell numbers were adjusted to an OD$_{600nm}$ of 0.2, and 5 µl of the cell suspension were spotted onto the indicated selective agar plates and incubated at 37°C for 3 d.

### In vitro binding assays

Pulldown assays using immobilized GST fusion proteins or biotinylated peptide (Biotin-CQGLYFHI NQTLREAHFHSLQHRG-COOH; PANATecs GmbH, Tübingen, Germany) were essentially performed as described (*Böhm et al., 2011*), using 10 µl glutathione or streptavidin sepharose beads, respectively, 0.76 nmole GST fusion or 5 nmole peptide, respectively, and 0.2 nmole (monomer) p97 or

0.5 nmole His$_6$-FAF1 in pulldown buffer (TBS containing 2 mM DTT and 1% or 0.1% Triton X-100 for glutathione and streptavidin pulldowns, respectively).

## Bioinformatic analyses

Sequence alignments were generated using the L-INS-I algorithm of the MAFFT package (*Katoh and Standley, 2013*). The multiple alignments were used for the generation of generalized profiles using pftools (*Bucher et al., 1996*). Generalized profile searches were performed using all proteins from the UniProt database (https://www.uniprot.org). Protein clustering was performed using the CLANS software (*Frickey and Lupas, 2004*). Sequence logos were generated using the WebLogo server (https://weblogo.berkeley.edu).

## Structural modeling

For structure prediction, the sequence of the p97 N domain (residues 1–213 of human p97) and the sequence of the VCF1 C-terminal region (residues 180–207 of human VCF1 isoform 1) were fed into a local installation of AlphaFold Multimer running locally with two GPUs (*Evans et al., 2022*). Five models were created, which showed no visually discernible differences for the interface in the central region, accompanied by a high AlphaFold confidence score around residue L191 of VCF1. A similar approach was used for the *Drosophila melanogaster* homologs, which created highly similar results.

## Mammalian cell culture

HeLa (ATCC; CCL-2) and HEK293T (ATCC; CRL-3216) cells and their derivatives were cultured in Dulbecco's modified Eagle's medium (DMEM) supplemented with 10% fetal bovine serum and 1% penicillin/streptomycin in a humidified atmosphere with 5% CO$_2$ at 37°C. In addition, 1.5 µg/ml puromycin or 400 ng/ml neomycin was added to the culture media for knockout or pINDUCER20-transduced cell pools, respectively. Mycoplasma contamination was excluded by performing PCR-based tests.

To create *VCF1/2* knockout cell pools, recombinant lentiviruses were produced using pLentiCRISPRv2-derived plasmids encoding gRNAs targeting the human *VCF1* and *VCF2* genes, respectively, generated using 3Cs technology (*Wegner et al., 2019*; gift from Manuel Kaulich, University of Frankfurt). The gRNA sequences used are listed in Appendix 1—key resources table. After production of recombinant lentiviruses in HEK293T cells by co-transfection of the pLentiCRISPRv2 constructs with pMD2.G and psPAX2 (Addgene plasmids #12259 and #12260; gifts from Didier Trono), HeLa and HEK293T cells at 40–60% confluence were transduced with lentivirus-containing supernatant mixed with polybrene to a final concentration of 8 µg/ml. Then, 48 hr after transduction, fresh culture medium containing 1.5 µg/ml puromycin was added, and the cell pools were kept under constant selection.

To generate HeLa cell pools ectopically expressing N-terminally FLAG epitope-tagged wild-type p97 and its variant carrying an N-terminal SV40-derived cNLS, pINDUCER20 with the respective inserts together with pMD2.G and psPAX2 was used to produce lentivirus-containing supernatant, and HeLa cells were transduced exactly as described above. Then, 48 hr after transduction, fresh culture medium containing 1000 ng/ml neomycin was added and selection was performed for 5 d. Expression of FLAG-tagged p97 variants was induced by addition of 1000 ng/ml doxycycline for 40 hr.

For the ectopic expression of VCF1/2, cells were transfected at 60% confluence using polyethylenimine and analyzed 42 hr after transfection. For siRNA-mediated depletion of VCF1, cells were seeded in 12-well plates and transfected with FAM104A or control siRNA (25 or 50 nM final concentration) using 1.2 ul Oligofectamine (Thermo Fisher) diluted in 100 ul Opti-MEM (Thermo Fisher). After 20 hr, the medium was changed and the knockdown was continued for a total of 72 hr.

## Immunoprecipitation experiments

For immunoprecipitation of FLAG-epitope-tagged VCF1/2 proteins, HEK293T cells were transfected with pCMV-Tag2B encoding the indicated VCF1/2 variants or with empty vector. Then, 48 hr after transfection, cells were harvested, washed twice with cold PBS (140 mM NaCl, 2.7 mM KCl, 10 mM Na$_2$HPO$_4$, 1.8 mM KH$_2$PO$_4$), and resuspended in 1000 µl of lysis buffer (50 mM Tris-HCl pH 7.6, 150 mM NaCl, 2 mM MgCl$_2$, 1% NP40, 10% glycerol, 1 mM DTT) containing protease inhibitors (1 mM PMSF, 1× Roche complete protease inhibitor mix). The cells were lysed on ice for 10 min, and

lysates were sonicated six times for 9 s in a Branson Sonifier using a microtip at 35% output control. Cell debris was removed by centrifugation (20,000 × *g*, 20 min, 4°C), and the protein concentration of the supernatants was determined with the Pierce BCA Protein Assay Kit (Thermo Fisher) and adjusted to equal input levels. Then, 50 ul were taken as input sample, mixed with fivefold concentrated SDS-PAGE sample buffer (250 mM Tris-HCl pH 6.8, 10% [w/v] SDS, 30% glycerol, 500 mM dithiothreitol, bromophenolblue) and heat-denatured. Then, 2 mg of the input were incubated overnight at 4°C with 50 µl FLAG M2 agarose beads on a rotating wheel. Beads were collected by centrifugation (1400 × *g*, 4 min, 4°C), and the supernatant was aspirated. The beads were washed twice with lysis buffer containing protease inhibitors, once with lysis buffer without inhibitors, and once with TBS (50 mM Tris-Cl, pH 7.5, 150 mM NaCl). The immunoprecipitates were heat-denatured on the beads in 1× SDS-PAGE sample buffer and further analyzed by immunoblotting.

For immunoprecipation of endogenous p97, HEK293T were transfected with pCMV-Tag2B encoding the indicated VCF1/2 variants or with empty vector. Then, 24 hr post transfection, cells were washed twice with PBS, harvested, and frozen. Frozen cell pellets were resuspended in lysis buffer containing protease inhibitors, incubated for 10 min on ice, and sonified exactly as described above. After centrifugation (20,000 × *g*, 15 min, 4°C) and adjustment of protein concentration, lysates were pre-cleared using 20 µl of Protein G sepharose beads (1 hr, 4°C, on a rotating wheel). After centrifugation (1200 rpm, 4 min, 4°C), the pre-cleared supernatants were transferred to fresh 1.5 ml reaction tubes, and 40 µl of the samples were collected as input sample. Pre-cleared lysates were either incubated with antibody against endogenous p97 (3 µg) or unspecific rabbit IgGs (3 µg) for 1 hr, 4°C on a rotating wheel, followed by addition of 25 µl Protein G sepharose beads for 2 hr at 4°C on a rotating wheel. The following steps were exactly as described above for the FLAG IP.

## Immunoblotting

Protein samples were separated by SDS-PAGE and transferred onto PVDF membrane (Millipore) by semi-dry blotting using 1× Tris-glycine buffer (192 mM glycine, 25 mM Tris base, pH 8.3) supplemented with 20% methanol. The membrane was blocked with 5% milk in TBST (50 mM Tris-HCl pH 7.5, 150 mM NaCl, 0.1% Tween 20) and incubated with the indicated primary antibody in blocking solution overnight at 4°C. The membrane was washed with TBST (3 × 10 min), incubated with HRP-conjugated secondary antibody (Dianova) diluted 1:7,500 in blocking solution for 1 hr at room temperature (RT), washed again with TBST (3 × 10 min), and incubated with Clarity Western ECL Substrate (Bio-Rad). Chemiluminescence signals were detected using the Gel Doc XR+system (Bio-Rad), and immunoblot images were processed with Image Lab (Bio-Rad).

## Immunofluorescence

Cells grown on coverslips to 60% confluence were washed twice with PBS, fixed using 3.7% formaldehyde in PBS for 12 min at RT, washed twice with cold PBS, permeabilized with 0.2% Triton X-100 in PBS for 10 min at RT, washed with PBS, and blocked by incubation with 2% BSA in PBS for 1 hr at RT. Cells were incubated with the indicated primary antibodies (diluted in 2% BSA in PBS) overnight at 4°C, washed for 15 min with PBS, and incubated with the appropriate fluorophore-coupled secondary antibodies for 2 hr at RT. Hoechst staining was performed by incubating the cells for 10 min in PBS containing 2.5 µg/ml Hoechst 33342. Following this, cells were washed for 15 min with PBS and rinsed with water. Coverslips were mounted for microscopy with ProLong Glass Antifade Mountant and sealed with nail polish.

## Microscopy and image processing

Confocal immunofluorescence microscopy was performed at the Imaging Core Facility (Biocenter, University of Würzburg) using a Leica TCS SP2 confocal microscope equipped with an acousto optical beam splitter. Images were acquired using a 63×/1.4 oil immersion objective and Leica confocal software. Where higher resolution was required, 2× digital zooming was applied. Single planes or Z stacks were acquired using Diode UV (405 nm), Ar (488 nm), and HeNe (561 nm) lasers with three PMTs set to 419–452 nm, 497–541 nm, and 595–650 nm, respectively. Image processing was performed using Fiji (*Schindelin et al., 2012*) and CellProfiler (*McQuin et al., 2018*). In CellProfiler, a pipeline was created to define nuclei via the Hoechst staining (thresholding method: minimum cross-entropy) and cells via the p97 staining (propagation from nuclei, thresholding method: otsu). The cytoplasm was defined by

subtracting nuclei from cells. Following this segmentation, intensities were measured in the nucleus and the cytoplasm, and their ratio was calculated.

To analyze larger cell numbers, immunofluorescence microscopy was performed on an Operetta High-Content Imaging System (20×/0.4, air, non-confocal, random field-of-view selection). Images were analyzed using Harmony High Content Imaging and Analysis Software in analogy to the Cell-Profiler analysis described above. Briefly, intensities were measured in the nuclei and cytoplasm after background subtraction, allowing for ratio calculations.

## Chromatin fractionation

To perform cellular fractionations, HEK293T cells were seeded in 10 cm dishes and transfected with pCMV-Tag2B encoding the indicated VCF1 isoforms or with empty vector. Then, 24 hr post transfection, cells were washed twice with cold PBS, harvested using cell scrapers, and centrifuged for 5 min at 1.300 rpm. Ten percent of the cell material was centrifuged separately, directly resuspended in 100 µl SDS-PAGE sample buffer, and heat-denatured. The remaining cell pellets (containing around $3 * 10^7$ cells) were resuspended in 600 µl hypotonic buffer (20 mM HEPES-KOH pH 8.0, 5 mM KCl, 1.5 mM MgCl$_2$, 1 mM DTT), incubated on ice for 20 min, and lysed using a 27-gauge needle (10 strokes). The cytoplasmatic fraction was separated by centrifugation (10 min, 16.000 rpm, 4°C) from the nuclei-containing pellet and transferred to a separate reaction tube. After protein concentration determination and volume adjustment, 50 µl were taken, mixed with fivefold concentrated SDS-PAGE sample buffer and heat-denatured. The pellet was resuspended in 400 µl nuclear extraction buffer (15 mM Tris-HCl pH 7.5, 1 mM EDTA, 0.4 M NaCl, 10% sucrose, 1 mM DTT) and incubated on ice for 30 min with occasional vortexing. The chromatin-containing insoluble fraction was removed by centrifugation (20.000 rpm, 20 min, 4°C), and the pellet was resuspended in 400 µl twofold concentrated SDS-PAGE sample buffer and heat-denatured. Then, 50 µl from the supernatant containing the soluble nucleoplasm was mixed with fivefold concentrated SDS-PAGE sample buffer and heat-denatured.

To release chromatin-bound proteins by benzonase treatment, HEK293T cells were harvested and lysed in lysis buffer (10 mM HEPES pH 7.9; 10 mM KCl; 1.5 mM MgCl$_2$; 0.34 M sucrose; 10% glycerol; 1 mM DTT) containing 0.1% TritonX-100 and 1× Roche complete protease inhibitor mix. After incubation on ice for 20 min and centrifugation (1300 × $g$, 4°C, 5 min), supernatant containing the cytoplasmatic fraction was separated from the nuclei-containing pellet. To receive the pure cytoplasmatic fraction, the supernatant was centrifuged at 20,000 × $g$ at 4°C for 15 min, and the supernatant was transferred to a new tube. The nuclei-containing pellet was gently washed in lysis buffer without detergent and subsequently resuspended in nuclear extraction buffer (3 mM EDTA; 0.2 mM EGTA; 1 mM DTT). Following rocking at 4°C for 40 min, the soluble nucleoplasm was separated from the chromatin containing pellet by centrifugation (1700 × $g$, 4°C, 5 min). The chromatin containing pellet was resuspended in lysis buffer containing 25 units benzonase and 1 mM CaCl$_2$, incubated for 90 min at 37°C, and centrifuged (20,000 × $g$, 4°C, 10 min). The supernatant containing the re-solubilized chromatin-bound fraction was separated from the insoluble pellet. Protein concentrations of all fractions were determined, and samples for immunoblot analysis were prepared by adding fivefold concentrated SDS-PAGE sample buffer.

## Growth assays

To analyze the sensitivity of HeLa control and VCF1/2 double-knockout cells toward inhibition of p97 and camptothecin, 1000 cells per cell line were seeded in CellCarrier-96 Ultra microplates. Then, 20 hr post seeding, half of the wells were treated with 1 uM camptothecin for 70 min. After removal of the inhibitor and two washing steps with PBS, cells in all wells were grown in the absence or presence of CB-5083 (0–200 nM) for further 7 d. Media containing the indicated CB-5083 concentrations were refreshed every 1.5 d. The experiment was performed in two biological replicates with three technical replicates each. After a total of 8 d, cells were washed with PBS, fixed using 3.7% formaldehyde in PBS for 12 min at RT, and washed twice with PBS. Permeabilization and Hoechst staining were performed by adding 0.2% Triton X-100 in PBS with 2.5 µg/ml Hoechst 33342 for 10 min in the dark. After washing the wells two times with PBS, images were taken with an Operetta CLS High-Content Imaging System with 20-fold magnification and analyzed using the Harmony High-Content and Imaging Analysis Software (PerkinElmer). A total of 26 image fields per well were acquired.

## Quantitative reverse transcription PCR (qRT-PCR)

RNA was isolated from HeLa cells seeded in 6-well plates ($0.25 * 10^6$ cells/well) using a NucleoSpin RNA kit (Macherey-Nagel, REF 740955) following the manufacturer's instructions. cDNA was generated from 1 µg total RNA using the Transcriptor High Fidelity cDNA synthesis kit (Roche, 5081963001) with oligo(dT) primer and diluted 1:40. qRT-PCR reactions were performed on a QuantStudio 5 instrument (Thermo Fisher) using PowerUp SYBR Green Master Mix (Thermo Fisher, A25741) and the primer pairs listed in Appendix 1—key resources table. The FAM104A_1 primer pair recognizes isoforms 1 and 2 of VCF1, whereas FAM104A_2 recognizes isoform 2 only. Each experiment was performed in at least three technical replicates per biological replicate. The house-keeping genes HPRT and PBGD were used for normalization, and relative expression levels were calculated according to the $2^{-\Delta\Delta cq}$ method using QuantStudio Design & Analysis Software (Thermo Fisher).

## Statistical testing

All statistical analyses and graphing were performed using GraphPad Prism 9. As described in the individual figure legends, Student's *t*-tests (two-tailed) or two-way factorial ANOVA (one- or two-way, corrected by Bonferroni's or Dunnett's multiple-comparisons test) were applied to the raw data. For the growth assays shown in *Figure 7*, relative cell viability was calculated by normalizing the cell number of each condition to the respective mean cell number of all replicates in the absence of CB-5083, separately for VCF1/2 double-knockout and control cells. After verifying the assumption of normal distribution using the D'Agostino–Pearson omnibus $K^2$ test and excluding a few extreme outliers (<3% of the dataset), ANOVA was performed. This analysis was followed up by Bonferroni's multiple-comparisons test, which corrected for multiple comparisons using statistical hypothesis testing. A p-value<0.05 was considered significant and indicated as follows: ns, not significant, *p<0.05, **p<0.01, ***p<0.001, **** p<0.0001.

## Acknowledgements

We thank Manuel Kaulich, Didier Trono, and Steve Elledge for providing plasmids; Monica Gotta and Elmar Wolf for providing the anti-UBXN2B and anti-MYC antibodies, respectively; the Imaging Core Facility (Biocenter, University of Würzburg) for support with confocal microscopy; and members of the Buchberger lab for critical reading of the manuscript. This work was funded by the Deutsche Forschungsgemeinschaft (DFG, German Research Foundation) through grants GRK2243/1+2 (to AB), 440766788 (INST 93/1023-1-FUGG; Operetta CLS system), and HBFG-133-612 (Leica SP2 confocal laser scanning microscope).

## Additional information

### Funding

| Funder | Grant reference number | Author |
| --- | --- | --- |
| German Research Foundation | 440766788; INST 93/1023-1-FUGG | Christina Schuelein-Voelk |
| German Research Foundation | GRK2243/1+2 | Alexander Buchberger |
| German Research Foundation | HBFG-133-612 | Christina Schuelein-Voelk |

The funders had no role in study design, data collection and interpretation, or the decision to submit the work for publication.

### Author contributions

Maria Körner, Resources, Investigation, Visualization, Methodology, Writing - review and editing; Susanne R Meyer, Gabriella Marincola, Resources, Investigation; Maximilian J Kern, Investigation; Clemens Grimm, Visualization; Christina Schuelein-Voelk, Resources, Methodology; Utz Fischer, Resources; Kay Hofmann, Investigation, Visualization; Alexander Buchberger, Conceptualization,

Resources, Supervision, Funding acquisition, Writing - original draft, Project administration, Writing - review and editing

**Author ORCIDs**
Gabriella Marincola  https://orcid.org/0000-0001-9227-6554
Utz Fischer  http://orcid.org/0000-0002-1465-6591
Kay Hofmann  http://orcid.org/0000-0002-2289-9083
Alexander Buchberger  http://orcid.org/0000-0002-2836-0820

**Decision letter and Author response**
Decision letter https://doi.org/10.7554/eLife.92409.sa1
Author response https://doi.org/10.7554/eLife.92409.sa2

## Additional files

**Supplementary files**
• Supplementary file 1. Summary of VCF1/2 yeast two-hybrid hits.
• MDAR checklist

**Data availability**
All data analysed during this study are included in the manuscript and supporting files; Source Data files accompany the respective figures.

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

# Appendix 1

## Appendix 1—key resources table

| Reagent type (species) or resource | Designation | Source or reference | Identifiers | Additional information |
|---|---|---|---|---|
| Strain, strain background (*Escherichia coli*) | XL1 Blue | Stratagene | Cat# 200249 | |
| Strain, strain background (*E. coli*) | XL10 Gold | Agilent | Cat# 200516-4 | |
| Strain, strain background (*E. coli*) | BL21(DE3) pRIL | Agilent | Cat# 230245 | |
| Strain, strain background (*Saccharomyces cerevisiae*) | PJ69-4A | *James et al., 1996* | N/A | |
| Cell line (*Homo sapiens*) | HeLa | ATCC | CCL-2 | |
| Cell line (*H. sapiens*) | HEK293T | ATCC | CRL-3216 | |
| Cell line (*H. sapiens*) | HeLa VCF1 knockout cell pool | This study | N/A | |
| Cell line (*H. sapiens*) | HeLa VCF1/2 double-knockout cell pool | This study | N/A | |
| Cell line (*H. sapiens*) | HeLa non-human-target control cell pool | This study | N/A | |
| Cell line (*H. sapiens*) | HeLa pINDUCER20 control cell pool | This study | N/A | |
| Cell line (*H. sapiens*) | HeLa pINDUCER20 p97 wildtype cell pool | This study | N/A | |
| Cell line (*H. sapiens*) | HeLa pINDUCER20 SV40NLS-p97 cell pool | This study | N/A | |
| Cell line (*H. sapiens*) | HEK293T VCF1 knockout cell pool | This study | N/A | |
| Cell line (*H. sapiens*) | HEK293T VCF1/2 double-knockout cell pool | This study | N/A | |
| Cell line (*H. sapiens*) | HEK293T non-human-target control cell pool | This study | N/A | See 'Materials and methods,' section 'Mammalian cell culture' |
| Antibody | Anti-alpha-tubulin (mouse monoclonal) | Sigma-Aldrich | Cat# T5168, RRID:AB_477579 | WB 1:2500 |
| Antibody | Anti-GAL4-TA (mouse monoclonal) | Santa Cruz | Sc-1663 | WB 1:500 |
| Antibody | Anti-FLAG (rabbit polyclonal) | Thermo Fisher Scientific | Cat# PA1-984B, RRID:AB_347227 | WB 1:2000 |
| Antibody | Anti-FLAG (mouse monoclonal) | Sigma-Aldrich | Cat# F1804 | WB 1:2000 IF 1:200 |
| Antibody | Anti-VCP (rabbit polyclonal) | Bethyl Laboratories | Cat# A300-589A, RRID:AB_495512 | WB 1:5000 IF 1:300 |
| Antibody | Anti-VCP (mouse monoclonal) | Santa Cruz | Cat# sc-57492, RRID:AB_793927 | IF 1:50 |
| Antibody | Anti-FAF1 (rabbit polyclonal) | Max Planck Institute of Biochemistry, animal house | AB65 | WB 1:10,000 |
| Antibody | Anti-UBXN7 (rabbit polyclonal) | Sigma-Aldrich | Cat# HPA049442 | WB 1:1000 |
| Antibody | Anti-NPL4 (rabbit polyclonal) | Sigma-Aldrich | Cat# HPA021560 | WB 1:1000 |
| Antibody | Anti-UFD1 (rabbit polyclonal) | Proteintech | Cat# 10615 | WB 1:2000 |
| Antibody | Anti-Ubiquityl-Histone H2B (rabbit monoclonal) | Cell Signaling | Cat# 5546 | WB 1:1000 |
| Antibody | Anti-UBXN2B (rabbit polyclonal) | *Lee et al., 2018* | N/A | WB 1:1000 |
| Antibody | Anti-UBXD1 (rabbit polyclonal) | Novus | NBP2-57653 | WB 1:1000 |
| Antibody | Anti-MCM7 (rabbit polyclonal) | Proteintech | 11225-1-AP | WB 1:1000 |

*Appendix 1 Continued on next page*

*Appendix 1 Continued*

| Reagent type (species) or resource | Designation | Source or reference | Identifiers | Additional information |
|---|---|---|---|---|
| Antibody | Anti-MYC (rabbit monoclonal) | Biocenter, University of Würzburg, made in-house | N/A | WB 1:2000 |
| Antibody | Anti-mouse IgG HRP (goat polyclonal) | Dianova (Jackson ImmunoResearch) | Cat# 115-035-003, RRID:AB_10015289 | 1:7500 |
| Antibody | Anti-rabbit IgG HRP (goat polyclonal) | Dianova (Jackson ImmunoResearch) | Cat# 111-035-045, RRID:AB_2337938 | 1:7500 |
| Antibody | Alexa Fluor 488 goat anti-rabbit IgG (H+L) | Thermo Fisher Scientific | Cat# A-11070, RRID:AB_142134 | 1:500 |
| Antibody | Alexa Fluor 488 goat anti-mouse IgG (H+L) | Thermo Fisher Scientific | Cat# A-11017, RRID:AB_143160 | 1:500 |
| Antibody | Alexa Fluor 594 goat anti-rabbit IgG (H+L) | Thermo Fisher Scientific | Cat# A-11072, RRID:AB_142057 | 1:500 |
| Antibody | Alexa Fluor 594 goat anti-mouse IgG (H+L) | Thermo Fisher Scientific | Cat# A-11020, RRID:AB_141974 | 1:500 |
| Recombinant DNA reagent | psPAX2 | Addgene | Cat# 12260 | |
| Recombinant DNA reagent | pMD2.G | Addgene | Cat# 12259 | |
| Recombinant DNA reagent | pCMV-Tag2B | Agilent | Cat# 211172 | |
| Recombinant DNA reagent | pCMV VCF1 isoform 1 | This study | pAB2225 | |
| Recombinant DNA reagent | pCMV VCF1 isoform 2 | This study | pAB2208 | |
| Recombinant DNA reagent | pCMV VCF1 isoform 5 | This study | pAB2223 | |
| Recombinant DNA reagent | pCMV VCF2 isoform 3 | This study | pAB2277 | |
| Recombinant DNA reagent | pCMV VCF1 isoform 1 Cdel26 | This study | pAB2280 | |
| Recombinant DNA reagent | pCMV VCF1 isoform 2 Cdel26 | This study | pAB2275 | |
| Recombinant DNA reagent | pCMV VCF1 isoform 5 Cdel26 | This study | pAB2276 | |
| Recombinant DNA reagent | pCMV VCF1 isoform 1 delNLS | This study | pAB2230 | |
| Recombinant DNA reagent | pCMV VCF1 isoform 2 delNLS | This study | pAB2231 | |
| Recombinant DNA reagent | pCMV VCF1 isoform 5 delNLS | This study | pAB2227 | See 'Materials and methods,' section 'Plasmids' |
| Recombinant DNA reagent | pGEX-4T1 | Cytiva | Cat# 28954549 | |

*Appendix 1 Continued on next page*

*Appendix 1 Continued*

| Reagent type (species) or resource | Designation | Source or reference | Identifiers | Additional information |
|---|---|---|---|---|
| Recombinant DNA reagent | pGEX-4T1 VCF1 isoform 1 | This study | pAB2212 | |
| Recombinant DNA reagent | pGEX-4T1 VCF1 isoform 2 | This study | pAB2195 | |
| Recombinant DNA reagent | pGEX-4T1 VCF1 isoform 5 | This study | pAB2213 | |
| Recombinant DNA reagent | pGEX-4T1 VCF2 isoform 3 | This study | pAB2298 | |
| Recombinant DNA reagent | pGEX-4T1 VCF1 isoform 1 Cdel26 | This study | pAB2295 | |
| Recombinant DNA reagent | pGEX-4T1 VCF1 isoform 2 Cdel26 | This study | pAB2296 | |
| Recombinant DNA reagent | pGEX-4T1 VCF1 isoform 5 Cdel26 | This study | pAB2297 | |
| Recombinant DNA reagent | pGEX-4T1 VCF2 isoform 3 Cdel26 | This study | pAB2299 | |
| Recombinant DNA reagent | pGEX-4T1 VCF1 isof. 5 NL->AA | This study | pAB3163 | |
| Recombinant DNA reagent | pGEX-4T1 VCF1 isof. 5 NL->RR | This study | pAB3162 | See 'Materials and methods,' section 'Plasmids' |
| Recombinant DNA reagent | mini-pRSETA | *Perrett et al., 1999* | N/A | |
| Recombinant DNA reagent | mini-pRSETA UBXN7 | This study | pAB2008 | See 'Materials and methods,' section 'Plasmids' |
| Recombinant DNA reagent | mini-pRSETA p47 | *Allen et al., 2006* | pAB356 | |
| Recombinant DNA reagent | pQE30 FAF1 | *Jensen et al., 2001* | N/A | |
| Recombinant DNA reagent | pET28a(+) UFD1 | *Fernández-Sáiz and Buchberger, 2010* | pAB425 | |
| Recombinant DNA reagent | pET21d NPL4 | *Fernández-Sáiz and Buchberger, 2010* | pAB1340 | |
| Recombinant DNA reagent | pProExHT p97 | *Fernández-Sáiz and Buchberger, 2010* | pAB1312 | |
| Recombinant DNA reagent | pProExHT p97 N domain | *Fernández-Sáiz and Buchberger, 2010* | pAB1342 | |
| Recombinant DNA reagent | pProExHT p97 ND1 | *Fernández-Sáiz and Buchberger, 2010* | pAB1343 | |
| Recombinant DNA reagent | pProExHT p97 ΔN | *Rothballer et al., 2007* | pAB749 | |
| Recombinant DNA reagent | pGAD-C1 | *James et al., 1996* | N/A | |
| Recombinant DNA reagent | pGAD-C1 VCF1 isoform 1 | This study | pAB2217 | |
| Recombinant DNA reagent | pGAD-C1 VCF1 isoform 2 | This study | pAB2205 | |
| Recombinant DNA reagent | pGAD-C1 VCF1 isoform 5 | This study | pAB2215 | |
| Recombinant DNA reagent | pGAD-C1 VCF2 isoform 3 | This study | pAB2233 | See 'Materials and methods,' section 'Plasmids' |
| Recombinant DNA reagent | pGBDU | *James et al., 1996* | N/A | |

*Appendix 1 Continued on next page*

*Appendix 1 Continued*

| Reagent type (species) or resource | Designation | Source or reference | Identifiers | Additional information |
|---|---|---|---|---|
| Recombinant DNA reagent | pGBDU p97 | This study | pAB1184 | |
| Recombinant DNA reagent | pGAD-C1 VCF1 isoform 5 Cdel4 | This study | pAB2312 | |
| Recombinant DNA reagent | pGAD-C1 VCF1 isoform 5 Cdel7 | This study | pAB2313 | |
| Recombinant DNA reagent | pGAD-C1 VCF1 isoform 5 Cdel13 | This study | pAB2314 | |
| Recombinant DNA reagent | pGAD-C1 VCF1 isoform 5 Cdel18 | This study | pAB2315 | |
| Recombinant DNA reagent | pGAD-C1 VCF1 isoform 5 Cdel26 | This study | pAB2235 | See 'Materials and methods,' section 'Plasmids' |
| Recombinant DNA reagent | pINDUCER20 | *Meerbrey et al., 2011* | pAB3012 | |
| Recombinant DNA reagent | pINDUCER20 p97 wildtype | This study | pAB3041 | |
| Recombinant DNA reagent | pINDUCER20 SV40NLS-p97 | This study | pAB3261 | See 'Materials and methods,' section 'Plasmids' |
| Recombinant DNA reagent | pLentiCRISPRv2 | Addgene | Cat# 52961 | |
| Recombinant DNA reagent | Non-human control sgRNAs in pLentiCRISPRv2 | Manuel Kaulich, University of Frankfurt | N/A | |
| Recombinant DNA reagent | VCF1 sgRNAs in pLentiCRISPRv2 | Manuel Kaulich, University of Frankfurt | N/A | |
| Recombinant DNA reagent | VCF2 sgRNAs in pLentiCRISPRv2 | Manuel Kaulich, University of Frankfurt | N/A | |
| Sequence-based reagent | ON-TARGETplus human FAM104A siRNA-SMARTpool | Dharmacon | Cat# L-015015-02-0005 | |
| Sequence-based reagent | ON-TARGETplus human UBXN2B siRNA-SMARTpool | Dharmacon | Cat# L-025945-01-0005 | |
| Sequence-based reagent | ON-TARGETplus non-targeting pool | Dharmacon | Cat# D-001810-10-05 | |
| Sequence-based reagent | FAM104A_1_fwd | This paper | qPCR primer | CTCCGTCCCAGGAAAAGGAG |
| Sequence-based reagent | FAM104A_1_rev | This paper | qPCR primer | AGGGTTTCTGCTACTTCTTTTGG |
| Sequence-based reagent | FAM104A_2_fwd | This paper | qPCR primer | TGGCAACGAAGAAGACAACC |
| Sequence-based reagent | FAM104A_2_rev | This paper | qPCR primer | TCACTGCCTGAAGACTCTGTG |
| Sequence-based reagent | HPRT_fwd | This paper | qPCR primer | TGGACAGGACTGAACGTCTTG |
| Sequence-based reagent | HPRT_rev | This paper | qPCR primer | CAGTCATAGGAATGGATCTATCAC |
| Sequence-based reagent | PBGD_fwd | This paper | qPCR primer | CCCTGGAGAAGAATGAAGTGG |
| Sequence-based reagent | PBGD_rev | This paper | qPCR primer | TTCTCTGGCAGGGTTTCTAGG |
| Sequence-based reagent | Fam104A1-KO-2-R_79 | This paper | gRNA | CGTAGCTTCCATCCGCCAGC |
| Sequence-based reagent | Fam104A1-KO-3-R_38 | This paper | gRNA | CCTCGGGCCTTGGCTCTCGC |
| Sequence-based reagent | Fam104A2-KO-1-R_190 | This paper | gRNA | TGTCCGGGCTATTGATGCTG |
| Sequence-based reagent | Fam104A2-KO-2-R_170 | This paper | gRNA | ACCGCGCAGAACGCTTTGTT |

*Appendix 1 Continued on next page*

*Appendix 1 Continued*

| Reagent type (species) or resource | Designation | Source or reference | Identifiers | Additional information |
|---|---|---|---|---|
| Sequence-based reagent | Fam104A3-KO-1-R_172 | This paper | gRNA | CTCCGCGAAGAGAGGGAACA |
| Sequence-based reagent | Fam104A3-KO-3-R_70 | This paper | gRNA | CCGAAACACAACCCCCTCTG |
| Sequence-based reagent | Fam104B1-KO-1-R_187 | This paper | gRNA | CTGTATCTTGAGAATCCTGA |
| Sequence-based reagent | Fam104B1-KO-2-R_18 | This paper | gRNA | CTTGCTCTCTCTGGGATATT |
| Sequence-based reagent | Fam104B1-KO-3-R_139 | This paper | gRNA | CATTAATATCCCAGAGAGAG |
| Sequence-based reagent | Fam104B2-KO-1-R_19 | This paper | gRNA | GATTGTTACTGAACCCGATG |
| Sequence-based reagent | Fam104B2-KO-2-R_85 | This paper | gRNA | GTTTTCATGGAGTGATAATG |
| Sequence-based reagent | Non-human-target-309-KO-1-R_156 | This paper | gRNA | AACATGACGTTCAAGATTGG |
| Sequence-based reagent | Non-human-target-365-KO-5-R_5 | This paper | gRNA | ACCACTGTTCTACGCGCAGG |
| Sequence-based reagent | Non-human-target-415-KO-2-R_24 | This paper | gRNA | TTGAACGGGCCGCGGAAGCG |
| Sequence-based reagent | Non-human-target-42-KO-15-R_115 | This paper | gRNA | CCCGCATGACACCGTCACTT |
| Peptide, recombinant protein | Biotin- CQGLYFHINQTLREAH FHSLQHRG-COOH | PANATecs GmbH | N/A | See 'Materials and methods,' section 'In vitro binding assays' |
| Commercial assay or kit | Pierce BCA Protein Assay Kit | Thermo Fisher Scientific | Cat# 23225 | |
| Commercial assay or kit | QuikChange XLII Mutagenesis Kit | Agilent | Cat# 200521 | |
| Commercial assay or kit | Transcriptor High Fidelity cDNA synthesis kit | Roche | Cat# 5081963001 | |
| Commercial assay or kit | NucleoSpin RNA kit | Macherey-Nagel | Cat# REF 740955 | |
| Chemical compound, drug | Camptothecin | Selleckchem | NSC-100880 | |
| Chemical compound, drug | CB-5083 | Selleckchem | Cat# S8101 | |
| Commercial assay or kit | Clarity Western ECL Substrate | Bio-Rad | Cat# 1705061 | |
| Chemical compound, drug | cOmplete, EDTA-free Protease Inhibitor Cocktail | Roche | Cat# 04693132001 | |
| Chemical compound, drug | Glutathione Sepharose 4 Fast Flow | Cytiva | Cat# 17513202 | |
| Chemical compound, drug | Ni-NTA Agarose | QIAGEN | Cat# 30230 | |
| Chemical compound, drug | Opti-MEM | Thermo Fisher Scientific | Cat# 31985602 | |
| Chemical compound, drug | Polybrene | Santa Cruz | Cat# sc-134220 | |
| Chemical compound, drug | Polyethylenimine (PEI) | Polysciences | Cat# 23966-1 | |
| Chemical compound, drug | ProLong Glass Antifade Mountant | Thermo Fisher Scientific | Cat# P36980 | |
| Chemical compound, drug | Anti FLAG-M2 affinity agarose beads | Sigma-Aldrich | Cat# A2220 | |
| Chemical compound, drug | Oligofectamine | Thermo Fisher Scientific | Cat# 12252011 | |
| Chemical compound, drug | Benzonase Nuclease | Sigma-Aldrich | Cat# 70664 | |

*Appendix 1 Continued on next page*

*Appendix 1 Continued*

| Reagent type (species) or resource | Designation | Source or reference | Identifiers | Additional information |
|---|---|---|---|---|
| Chemical compound, drug | PowerUp SYBR Green Master Mix | Thermo Fisher | Cat# A25741 | |
| Chemical compound, drug | Protein G Sepharose 4 fast flow | Cytiva | Cat# 17-5132-01 | |
| Software, algorithm | Fiji | http://fiji.sc | RRID:SCR_002285 | |
| Software, algorithm | Image Lab Software | Bio-Rad | RRID:SCR_014210 | |
| Software, algorithm | CellProfiler | https://cellprofiler.org/ | RRID:SCR_007358 | |
| Software, algorithm | GraphPad Prism | https://www.graphpad.com/scientific-software/prism/ | RRID:SCR_002798 | |
| Software, algorithm | QuantStudio Design & Analysis Software | https://www.thermofisher.com/de/de/home/global/forms/life-science/quantstudio-3-5-software.html | | |

