## [Editor Report]

This article reports on hitherto unrecognized adaptors of p97/VCP, which is a multifunctional ATPase that unwinds diverse protein substrates subserving important roles in cell physiology. The adaptors in question, members of the FAM104 family, direct p97 to the nucleus to enable unwinding events in that location. The findings, which are supported by solid experimental observations, are valuable and will inform the work of the sizable community that studies various aspects of p97/VCP.

---

## [Decision Letter]

[Editors' note: this paper was reviewed by Review Commons.]

---

## [Author Response]

1. General Statements

We thank all three Reviewers for their constructive and encouraging feedback. We addressed all their comments (see point-by-point response below) to significantly improve the revised version of our manuscript.

Specifically, we

performed quantifications and statistical testing of all microscopy data, as well as high content microscopy analysis of p97 localization, confirming that loss of FAM104 proteins results in a statistically highly significant reduction in nuclear p97 levels (new Figures 6CDF; Figure 6—figure supplement 1CD), whereas ectopic expression of FAM104 proteins induces a statistically highly significant nuclear accumulation of p97 (new Figure 4C).showed that the fusion of the strong SV40 NLS to the N terminus of p97 can fully compensate for the loss of FAM104 proteins with respect to p97 nuclear localization (new Figure 6EF)performed additional control experiments demonstrating that the chromatin association of p97 induced by FAM104 proteins (i) depends on their NLS (new Figure 5—figure supplement 1A), and (ii) does not represent unspecific precipitation of p97 (new Figure 5B)showed that FAM104 proteins also bind to the p97-UBXN2B complex implicated in protein phosphatase 1 regulation (Figures 3A; Figure 3—figure supplement 1A), indicating that they likely control p97 functions beyond proteasomal degradation pathwaysfound that chemical inhibition of p97 and loss of FAM104 proteins possess additive effects on the nuclear localization of p97 (new Figure 7EF), thus explaining their observed synthetic growth defects.

Please note that the official gene names of FAM104A/B were changed to VCP nuclear cofactor family member 1 and 2 (VCF1/2) upon request by an editor of the HUGO Gene Nomenclature Committee (HGNC), after we had uploaded the preprint of our manuscript to bioRxiv. We therefore had to perform multiple adjustments to the manuscript text in order to adhere to the new nomenclature.

We also reorganized previous Figure S4 into new figure supplements to Figures 4 through 6, in order to better adjust the figures to the progression of the Results section.

Below, please find a point-by-point response to the Reviewers' comments.

2. Point-by-point description of the revisionsReviewer #1 (Evidence, reproducibility and clarity (Required)):[…]Major comments:- Are the claims and the conclusions supported by the data or do they require additional experiments or analyses to support them?Claims concerning the mapping of FAM104 and p97 (Figures 1&2) are generally well concluded. Yet, minor issues concerning Fig2D (FAM104Aiso1 cdel26) as well as Figure 2E (p97-deltaN pull down) lack of interaction-are not supported by the presented data (both show weak interactions).

We actually did not claim that Figures 2DE show a *complete* lack of interaction. Rather, we stated that "deletion of the C-terminal 26 residues *impaired* the ability of all four FAM104A/B isoforms to bind p97 in a pulldown experiment (Figure 2D)", and "We also noted some residual binding of the p97 ΔN variant, suggesting that the FAM104A-derived peptide has some weak affinity for a p97 region(s) outside the N domain." (Figure 2E). However, in light of the other in vitro binding data and the complete loss of p97 binding by the Cdel26 variants in the immunoprecipitation experiment (Figure 3A), we strongly feel that our claims regarding the binding of FAM104 proteins via their C-terminal helix to p97 are well supported.

Claims concerning nuclear/cytosol p97 distribution impact upon FAM104 manipulations (over-expression or KO) need to be further evaluated by additional methodologies. For example, the distribution impact using the FAM104 mutants in 4B should be evaluated by cell fractionation experiments (as performed in figure 5). Cell fractionation performed for FAM104A isoforms 1 and 2 should be performed on isoforms 5 and 3, the fact that they are expressed at lower levels has no impact, as the evaluation is on p97 and they were able to show in figure 3A an impact on p97 levels. Impact on distribution performed in Figure 6 using FAM104 KO cells should also include cell fractionation experiments in order to enable clear conclusion regarding FAM104 impact on p97 nuclear distribution.

Compared to VCF1 isoforms 1 and 2, wild-type VCF1 isoform5 and VCF2 isoform3 are expressed at much lower levels (an estimated 30 and 100 times less, respectively), as evident from the inputs shown in Figure 3A and new Figure 3–figure supplement 1AC. Importantly, reduced levels of soluble p97 were only observed for the strongly overexpressed VCF1 isoforms 1 and 2 (see Figures 3A and Figure 3–figure supplement 1AC, inputs), clearly indicating that the expression levels correlate with the impact on p97 solubility. Given the low expression levels of the other two isoforms, a reduction of the highly abundant p97 protein in fractionation experiments cannot be expected. We nevertheless performed the requested experiment (see Author response image 1). Expression of VCF1 isoform 5 was much weaker (30-40 times less) than that of isoforms 1 and 2, requiring very long exposure times for detection, and did not result in any detectable redistribution or reduction of p97 compared to the vector control. The even more weakly expressed VCF2 isoform 3 could not be detected at all after fractionation.

**Author response image 1. sa2fig1:** Fractionation of cell lysates after ectopic expression or deletion of VCF1/2. (A) HEK293T cells ectopically expressing the indicated VCF1 isoforms were processed to cytoplasmic, soluble nuclear (nucleoplasmic) and chromatin fractions and analyzed by immunoblotting as indicated. (B) As in (A), but using VCF1 single and VCF1/2 double knockout cell pools transfected with siRNA targeting VCF1. (C) Quantification of the ratio of nucleoplasmic to cytoplasmic p97 levels in (B). p97 band intensities were normalized to the whole lane intensity of the total protein stain (Coomassie) of the respective lane. The p97 intensities in the nucleoplasmic fraction were divided by the intensity in the cytoplasmic fraction, and this ratio was set to 1 for the control cells.

As requested, we performed the fractionation experiments with cells expressing the Cdel26 and delNLS mutants of VCF1 isoforms 1 and 2 used in Figure 4B. Clearly, expression of the p97 binding-deficient Cdel26 variants did not result in any depletion of cytoplasmic p97 or accumulation of chromatin-bound p97 (new Figure 5—figure supplement 1A), in good agreement with the microscopy data shown in Figure 4B. Expression of the NLS-deleted variants resulted in some accumulation of p97 in the nuclear fractions, consistent with the nuclear/chromatin localization of these mutant VCF1 proteins (new Figure 5—figure supplement 1B).

We also performed the requested fractionation of VCF1 KO and VCF1/2 double KO cell lysates after additional siRNA-mediated depletion to reduce any residual VCF1 expression (Author response image 1; see Figure 7—figure supplement 2 for residual VCF1 expression in the knockout cell pools). Even though not obvious to the naked eye, normalization revealed a reduction in the ratio of nucleoplasmic to cytoplasmic p97 (Author response image 1). However, such reduction was not always observed, potentially because the fractionation experiments are not sensitive enough to reliably detect differences in the range of 25% of the nuclear/cytoplasmic ratio. We therefore felt uncomfortable including these data in the revised manuscript. Instead, we corroborated the microscopy data shown in Figure 6 by performing high content microscopy followed by automated image analysis of more than 3,000 cells each in five biological replicates per cell pool (in total, >15,000 cells per cell pool), which revealed a highly significant reduction by 25% of the nuclear/cytoplasmic p97 ratio in the VCF1/2 double KO cells (new Figure 6—figure supplement 1D), in excellent agreement with the previous confocal microscopy analyses. This result was further confirmed in the high content microscopic analysis shown in new Figure 6EF, where VCF1-depleted cells under control conditions consistently showed a significantly reduced nuclear/cytoplasmic p97 ratio. Importantly, this reduction was eliminated upon ectopic expression of a cNLS-p97 fusion protein, in line with the hypothesis that VCF1/2 binding provides p97 with a strong additional cNLS promoting its efficient nuclear import. Taken together, the impact of VCF1/2 on the nuclear localization of p97 was consistently observed upon overexpression, CRISPR/Cas-mediated genomic deletion and siRNA-mediated de(p)letion, in different cell lines and under different experimental conditions.

Also, statistics presented are somewhat problematic at several points. In figure S4C the ** difference between vector and deNLS mutant make no sense (I think they should have been non-significant). Figure 7 make no sense to compare WT and KO cells (in panels B and C) if their original growth was different. One should compare the differences in respect to the drug concentration in each cell type. Also, it may be useful for statistical purposes to evaluate cell numbers rather than growth% and this may enable to obtain better statistical significances.

Upon request by Reviewers #1 and #3, we re-analyzed the microscopy data shown in Figures 4AB using an improved (log transformed) quantification of cytoplasmic pixels that allows for a more robust detection of weak cytoplasmic signals and show the quantifications in new Figures 4C and 4D. One-way ANOVA confirmed that the strong nuclear enrichment of p97 upon ectopic expression of all four full-length VCF1/2 proteins is highly statistically significant, whereas expression of NLS-deleted or p97 binding-deficient variants does not lead to a significant change in p97 localization in comparison to the vector control (new Figure 4C).

Regarding Figure 7BC, we actually did account for the growth difference of the WT and KO cells by normalizing growth separately for the WT and KO cells to the respective mean values at 0 nM CB-5083, as stated in the figure legend. We also explained this in the main text: "Note that growth of both cell lines was normalized to their respective growth in the absence of CB-5083, thus taking account of the slower growth of the knockout cells under unperturbed conditions." Thus, the growth differences for the various drug concentrations are shown relative to the respective cell type, as requested by the reviewer. To do so, the absolute cell numbers for each condition were divided by the mean of the cell numbers in the absence of CB-5083 for the same cell type, resulting in relative growth (in %). These numbers are thus directly derived from cell numbers. In order to avoid confusion with growth rates, we renamed the y-axis in Figures 7BC to "relative cell count (%)".

– Are the data and the methods presented in such a way that they can be reproduced? Y2H is not explained at all in methods, furthermore, it would be useful to present in a table the entire list of p97 interactome obtained in this screen.

We apologize for not including information on the Y2H experiments beyond the cloning procedures in the Methods section. (Please note that some experimental detail was given in the legend to Figure 1C.) We added a section explaining the Y2H assays to the Methods section of the revised manuscript.

We are currently following up several other candidates from the Y2H screen that are unrelated to the FAM104 story. We therefore decided to refrain from disclosing the identities of those candidates at this point. However, we added a table giving a detailed description of all VCF1/2 hits from our screen as new Supplementary Table 1.

Are prior studies referenced appropriately?Previous reports regarding FAM104 interaction with p97 have been reported in two papers (PMID 32296183 sup. Table9 therein and PMID 32814053 S2 therein) this has not been stated at all. Furthermore, no data concerning previous knowledge of FAM104 is referred to in the introduction.

We actually cited several publications reporting the interaction between VCF1/FAM104A and p97 in the second paragraph of the Discussion, including PMID 32814053 (Haenig et al., 2020). We did not cite PMID 32296183 (Luck et al., 2020), because it is essentially an extension of PMID 25416956 (Rolland et al., 2014), who first reported the Y2H interaction between FAM104A and p97 (and whom we cited). For the sake of completeness, we included the Luck et al. reference in the revised manuscript.

We decided to summarize previously published information on FAM104A in the Discussion rather than the Introduction, because all information comes from high-throughput studies and had never been validated in follow-up experiments before; with the exception of the supposed co-IP with p47 (Raman et al., 2015), which we were unable to confirm (see Figure 3—figure supplement 1A).

Referee cross-commentingIt seems reviewer #2 concerns are also situated close to our comments regarding nuclear function of FAM104 on p97 function. Reviewers 3 comment regarding UBXN2B possible tertiary complex with p97 and FAM104 should be attempted as it would help put p97 function in a slightly more specific context.

As requested by Reviewers #1 and #3, we blotted the FLAG-VCF1/2 IPs against UBXN2B and could indeed confirm the co-IP of UBXN2B with all four VCF1/2 tested (Figures 3A; Figure 3—figure supplement 1A). The levels of co-precipitated UBXN2B correlated with those of p97, and the co-IP of UBXN2B was lost with the p97 binding-deficient Cdel26 variants of VCF1, indicating that VCF1/2 and UBXN2B indeed form a ternary complex with p97, in contrast to the closely related cofactor p47. These data show that VCF1/2 can associate with p97 complexes other than p97-UFD1-NPL4, but not with all p97 complexes. The specific impact of VCF1/2 on p97-UBXN2B complexes remains to be addressed in future work.

[…]Reviewer #2 (Evidence, reproducibility and clarity (Required)):[…]

We thank the reviewer for the detailed summary of the paper and positive comments about the work.

The authors hypothesize that FAM104 proteins enhance the nuclear/chromatin-associated function of p97/VCP by sequestering it from the cytosol into nuclear/chromatin. In the corresponding experiments, overexpression of FAM104 species (Figures 4 and 5) in otherwise unperturbed cells is used. Because recruitment of p97/VCP to client proteins is thought to depend in large part on ubiquitylation, it is unclear how overexpression of FAM104 is sufficient to enhance nuclear/chromatin localization of VCP. Is nuclear/chromatin localization accompanied by changes in ubiquitylation and/or turnover of the corresponding proteins? In other words, does enhanced localization also correlate with increased activity, or could the enhanced nuclear/chromatin association also be explained by inhibited/captured p97/VCP?

Our data are consistent with the simple model that the FAM104 proteins VCF1/2 bind strongly to p97 and promote its nuclear import via their potent cNLS. In addition, VCF1 isoform 2 and particularly isoform 1 associate with nuclear chromatin independently of p97 (new Figure 5—figure supplement 1A), suggesting that their affinity for chromatin drives the sequestration of p97 on chromatin. Thus, the observed effects on p97 localization appear to be the consequence of VCF1/2 protein localization and are therefore in the first instance independent of client proteins and their ubiquitylation state. As stated in the Discussion, FAM104 proteins in that respect resemble two other small p97 interactors combining a helical p97 binding motif with a localization feature, i.e. SVIP and VIMP, which recruit p97 to endo/lysosomes and the ER, respectively.

We also determined the effect of VCF1/2 overexpression on the co-IP of other cofactors and substrates with endogenous p97 (new Figure 3—figure supplement 1C) and found that the strongly overexpressed VCF1 isoforms 1 and 2 efficiently outcompeted the major cofactors UFD1-NPL4 and p47 as well as bona fide polyubiquitylated substrates, while the much weaker expressed VCF1 isoform 5 and VCF2 isoform 3 did not interfere with cofactor binding and actually increased the association of ubiquitylated substrates with p97. In light of these results, we cannot exclude the possibility that the strong chromatin association of p97 seen upon ectopic expression of VCF1 isoforms 1 and 2 is caused by inhibited/captured p97. However, the much weaker ectopic expression of VCF1 isoform 5 and VCF2 isoform 3 clearly induces the nuclear accumulation of p97 (Figure 4AC) in the absence of adverse effects on p97 cofactor binding (new Figure 3—figure supplement 1C). Moreover, de(p)letion of VCF1/2 clearly reduced the ratio of nuclear/cytoplasmic p97, indicating that endogenous VCF1/2 indeed promote the nuclear localization of p97.

The authors link the function of FAM104 proteins in nuclear targeting of p97/VCP to the absence of a unique NLS peptide. Therefore, it would be interesting to determine whether the appearance of FAM104 proteins at the evolutionary level correlates with the strength/presence of NLS peptides in p97/VCP and/or its cofactors UFD1/NPL4/FAF1/UBXN3. Do FAM104 proteins compensate for the loss of NLS peptides in p97/cofactor complexes?

In new Figure 1—figure supplement 1C, we show multiple sequence alignments for the N-terminal region of p97 homologs of species possessing or lacking FAM104 homologs. All p97 homologs from yeast to mammals possess a relatively weak potential cNLS (K60R65 in human p97, score 5 according to cNLS mapper). Available three-dimensional structures of p97 suggest that this potential cNLS is part of a β-hairpin in the N domain that is unlikely to be constitutively exposed. In yeast Cdc48, an additional bi-partite cNLS closer to the N terminus has been experimentally validated (Madeo et al., MBoC 1998). Interestingly, this additional NLS is conserved in several p97 homologs from species lacking FAM104 proteins, but lost or poorly conserved in most p97 homologs from species possessing FAM104 proteins (new Figure 1—figure supplement 1C), consistent with the hypothesis that FAM104 proteins compensate for the loss of the functional bipartite cNLS in those p97 homologs. We obtained further experimental support for this hypothesis by fusing an efficient NLS peptide to p97 – see reply to point (3) below.

Analysis of potential NLS peptides in UFD1, NPL4, FAF1 and UBXN7 using cNLS mapper revealed a strongly predicted cNLS for UBXN7, which is not conserved in the yeast homolog Ubx5, but only weakly predicted or no cNLSs for the other cofactors. We feel that a more detailed bioinformatic analysis of the evolutionary conservation of potential NLSs in a large number of p97 cofactors is beyond the scope of the present study.

Re (2) It remains unclear whether FAM104 proteins are responsible for the mere sequestration of p97/VCP in the nucleus or whether FAM104 proteins also contribute to process/client specificity in other ways. In this context, the authors could investigate a possible compensation of the reduced nuclear targeting of p97/VCP in FAM104 knock-out cells by fusion with an efficient cNLS peptide. Does this compensate for both nuclear/chromatin localization and growth/drug sensitivity?

We indeed favor the idea that FAM104 proteins merely promote the nuclear localization of p97. Upon request by Reviewers #1 and #3, we blotted the FLAG-VCF1/2 IPs against UBXN2B and found a robust co-IP of UBXN2B with all four VCF1/2 proteins tested (Figures 3A; Figure 3—figure supplement 1A). The levels of co-precipitated UBXN2B correlated with those of p97, and the co-IP of UBXN2B was lost with the p97 binding-deficient Cdel26 variants of VCF1, indicating that FAM104 proteins and UBXN2B form a ternary complex with p97, in contrast to the closely related cofactor p47. These data show that VCF1/2 can associate with p97 complexes other than p97-UFD1-NPL4, arguing against a narrow process/client specificity of VCF1/2-containing p97 complexes.

We thank the reviewer for the excellent suggestion regarding the cNLS-p97 fusion protein. To address this point, we fused the prototypical cNLS of SV40 to the N terminus of p97 and analyzed the effect on p97 localization and on cell growth. In the high content microscopic analysis shown in new Figure 6EF, VCF1-depleted cells showed a significantly reduced nuclear/cytoplasmic p97 ratio under control conditions. Importantly, this reduction was eliminated upon doxycycline-induced ectopic expression of the cNLS-p97 fusion protein, showing that the cNLS fusion can indeed compensate for the lack of VCF1/2. Unfortunately, we could not observe a similar compensation of the growth/drug sensitivity of VCF1/2 knockout cells (data not shown). It is possible that the stronger nuclear accumulation of p97 among expression of the cNLS-p97 fusion protein compared to control cells has adverse effects on cell growth, or that the N-terminal cNLS fusion interferes with normal p97 activity, among other possibilities.

Re (3) How does overexpression of FAM104 alter drug sensitivity compared to knock-out cell lines (Figure 7)?

To address this point, we transfected a plasmid encoding the strongly expressed VCF1 isoform 1 into HEK293T cells, which can be more efficiently transfected than HeLa cells. We detected weakly reduced growth and slightly enhanced CB-5083 sensitivity compared to control cells 24h post transfection (new Figure 7—figure supplement 1). The true effect is probably underestimated because the transfection efficiency was around 50 – 60%. The observed negative effect on cellular fitness is consistent with the massive impact of VCF1 isoform1 overexpression on cofactor binding to p97 (new Figure 3—figure supplement 1C; see also reply to point 1 above), strongly suggesting that a balanced expression of FAM104 proteins is crucial for optimal p97 function.

Is there experimental evidence on how FAM104 proteins can bind p97/VCP to chromatin in this context and the proposed targeting of p97/VCP to the nucleus/chromatin? Does FAM104 mRNA/protein expression increase when p97/VCP-mediated processes are disrupted (e.g., in the presence of p97/VCP inhibition or DNA damage)? Are FAM104 protein levels stabilized under these conditions? Are FAM104 proteins differentially regulated (e.g., in terms of localization) under these conditions? Figure 3A suggests that FAM104 proteins may have a different function in relation to p97/VCP protein levels: FAM104A iso1/2 have lower p97/VCP protein levels than FAM104A iso5 and B iso3. The authors suggest that this is due to the solubility of p97/VCP. It should be clarified whether lower solubility equates to increased chromatin association.

Regarding the mechanism of p97 targeting to the nucleus and to chromatin, please see the first part of our reply to point 1 above.

Regarding VCF1/2 expression levels and regulation, we unfortunately were unable to analyze this on the protein level, because commercial antibodies detecting endogenous VCF1 or 2 are not available. (Please note that the antibodies available via the Human Proteome Atlas project detect either proteins of the wrong size (VCF2/FAM104B) or require ectopic expression of the specific target (VCF1/FAM104A).) Nevertheless, we were able to identify PCR primer pairs enabling the efficient and specific amplification of VCF1 isoform 1 and 2 cDNA. RT-qPCR analysis revealed that CB-5083 treatment indeed induced an up to two-fold increase in VCF1 expression (new Figure 7D), in line with an increased demand for VCF1/2 under conditions of limiting p97 activity. Intriguingly, the microscopic analysis shown in new Figure 7EF demonstrates that not only VCF1/2 deletion, but also CB-5083 treatment reduces nuclear p97 levels, and that both conditions have an additive effect, thereby providing a straightforward explanation for the synthetic growth defects observed in Figure 7BC.

Regarding the effect of ectopic expression of VCF1 isoforms 1 and 2 on the solubility of p97, we used an alternative fractionation protocol including a benzonase treatment step to solubilize chromatin-bound proteins. Comparing cells overexpressing VCF1 isoform 1 with control cells, we found that the majority of p97 is re-solubilized upon benzonase treatment, similar to VCF1 isoform1 itself and the chromatin-associated proteins MYC, MCM7 and UbH2B (new Figure 5B). We therefore conclude that the lower solubility of p97 upon overexpression of VCF1 isoform 1 (and, by extension, isoform 2) indeed reflects increased chromatin association. While we believe that the differences between VCF1 isoforms 1/2 versus VCF1 isoform 5 and VCF2 isoform 3 observed in the fractionation and IP experiments reflect the very strong differences in expression levels rather than intrinsically different functions in relation to p97, this interesting point remains to be addressed in future work.

It remains unclear whether a FAM104-dependent shift in nuclear/chromatin-associated p97/VCP could also be a secondary compensatory effect versus functional impairment in FAM104 overexpression/de(p)letion. The authors might include this in their discussion.

We agree that the ectopic expression of VCF1/2 could, in principle, promote the nuclear localization indirectly, e.g. by competing out other p97 cofactors and thereby inducing partial loss of function of p97 leading to its nuclear accumulation. We actually believe that the very strong accumulation of p97 on chromatin upon ectopic expression of VCF1 isoforms 1 and 2 (Figure 5) is likely to be unphysiological in light of their very high expression levels and of their massive impact on cofactor binding to p97 (new Figure 3—figure supplement 1C). However, we would like to emphasize that the nuclear (not chromatin) accumulation of p97 was also observed for the much more weakly expressed VCF1 isoform 5 and VCF2 isoform 3 under conditions of unperturbed p97-cofactor interactions (new Figure 3—figure supplement 1C), strongly suggesting that the nuclear accumulation of p97 is the primary result of its tight binding to the nuclear VCF1/2 proteins. Even more importantly, the reduction in nuclear p97 upon de(p)letion of VCF1/2 is extremely unlikely to be a secondary effect, as one would have to postulate that loss of the nuclear VCF1/2 proteins would lead to a retention of p97 in the cytoplasm by some unclear mechanism. We think that the accumulated evidence strongly supports a direct regulatory function of VCF1/2 on the nuclear localization of p97 and is hardly compatible with the scenario that the observed relocalizations of p97 are merely secondary, compensatory effects of increased or reduced VCF1/2 levels, respectively. As suggested, we added a short passage addressing this point to the Discussion section.

Reviewer #2 (Significance (Required)):In summary, the author's conclusion that FAM104 proteins represent a previously underappreciated class of p97/VCP cofactors is well supported. Given the versatile and important role of p97/VCP and cellular protein homeostasis pathways, this finding is of interest to a broad audience. However, the functional role of FAM104 proteins in p97/VCP biology remains unclear. Therefore, the authors need to further elaborate the physiological contribution of FAM104 proteins to p97/VCP function in additional experiments. The suggestions are largely based on modifications of experiments already performed in this manuscript.

We thank the reviewer for appreciating the broad relevance of our findings. We performed additional experiments to characterize the synthetic defects of VCF1/2 knockout and p97 inhibition (new Figure 7EF). Importantly, we found that inhibitor treatment alone reduced nuclear p97 levels, and that loss of VCF1/2 led to a further decrease. These results indicate that the synthetic growth defects observed in Figure 7BC are caused by the even stronger reduction of nuclear p97 to an extent that is no longer compatible with viability. At the same time, we don´t think that VCF1/2 possess highly specific functions in one or few p97-dependent pathways, which is in line with our finding that VCF1/2 can not only bind to p97-UFD1-NPL4, but also to the p97-UBXN2B complex possessing distinct functions (Figures 3A; Figure 3—figure supplement 1A). In essence, we believe that the nuclear targeting of p97 in fact *is* the physiological function of VCF1/2, in analogy to similar small proteins that were recently shown to promote the nuclear import of the 20S proteasome and HUWE1. We added a section addressing the emerging general concept of such small NLS-providing cofactors to the Discussion.

Reviewer #3 (Evidence, reproducibility and clarity (Required)):[…]

We thank the reviewer for the positive comments about the work.

P3: The authors write that mutations in VCP are causative for cancer. This should be rephrased.

We actually did not write that mutations in VCP are causative for cancer. Rather, we stated that mutations in VCP are causative for MSP1, and that "several cancers and viruses rely on p97 activity", which we believe is correct.

P3: I would suggest to add a reference to the new study that also shows that VCP is also exploited by bacteria rand not only viruses.

We thank the referee for this suggestion. We now state that "viral and bacterial pathogens rely on p97 activity" and added a reference.

Could the authors better illustrate the difference between FAM104A and B, and provide some explanations of why A seems to interact better with VCP compared to B. Is it just matter of higher expression of FAM104A in the cells where the interaction has been tested?

We slightly adjusted the color scheme of Figure 1A and added the information that the central regions of FAM104 proteins possess (lower) sequence homology in the Results section. Based on the in vitro pulldown experiments, we think that the VCF1 and VCF2 isoforms bind to p97 with similar affinities. The apparent differences between VCF1 and VCF2 seen in the immunoprecipitation experiments are indeed most likely a consequence of strong differences in expression levels, as can be appreciated e.g. in the input samples in Figures 3A and Figure 3—figure supplement 1AC.

The author should quantify the IF results in Figure 4 and include the quantification in the main figure.

We quantified the results as requested using an improved (log transformed) quantification of cytoplasmic pixels that allows for a more robust detection of weak cytoplasmic signals and show the quantifications in new Figures 4C and D. One-way ANOVA confirmed that the strong nuclear enrichment of p97 upon expression of all four full-length VCF1/2 proteins is highly statistically significant, whereas expression of NLS-deleted or p97 binding-deficient variants does not lead to a significant change in p97 localization (new Figure 4C). The quantification of the FLAG signals showed that the full-length and p97 binding-deficient VCF1/2 proteins are strongly enriched in the nucleus, whereas the NLS-deleted variants are not (new Figure 4D).

UBXN2B interaction with FAM104A was found in HT affinity-MS (Huttlin et al) and Y2H (Luck et al) studies. Can the authors validate this interaction of UBXN2B with FAM104 proteins? This would help to understand whether FAM104 interacts mainly with nuclear adaptors.

As requested by Reviewers #1 and #3, we blotted the FLAG-VCF1/2 IPs against UBXN2B/p37 and could indeed confirm the co-IP of UBXN2B with all four VCF1/2 proteins tested (Figures 3A; Figure 3—figure supplement 1A). The levels of co-precipitated UBXN2B correlated with those of p97, and the co-IP of UBXN2B was lost with the p97 binding-deficient Cdel26 variants of VCF1, indicating that VCF1/2 and UBXN2B indeed form a ternary complex with p97, in contrast to the closely related cofactor p47. These data show that VCF1/2 can associate with p97 complexes other than p97-UFD1-NPL4, but not with all p97 complexes. The specific impact of VCF1/2 on p97-UBXN2B complexes remains to be addressed in future work. Please note that our results do not allow conclusions about the subcellular localization of the ternary p97-UBXN2B-VCF1/2 complexes, because endogenous as well as ectopically expressed UBXN2B has been shown to localize to the cytoplasm and to the nucleus (Uchiyama et al., Dev Cell 2006; Raman et al., Nat Cell Biol 2015).